# Last-iterate Convergence Separation between Extra-gradient and Optimism in Constrained Periodic Games

**Yi Feng**[1]      **Ping Li**[1]      **Ioannis Panageas**[2]      **Xiao Wang**[1,3]

[1]Institute for Theoretical Computer Science, Shanghai University of Finance and Economics
[2]Department of Computer Science, University of California Irvine
[3]Key Laboratory of Interdisciplinary Research of Computation and Economics, Ministry of Education

## Abstract

Last-iterate behaviors of learning algorithms in repeated two-player zero-sum games have been extensively studied due to their wide applications in machine learning and related tasks. Typical algorithms that exhibit the last-iterate convergence property include optimistic and extra-gradient methods. However, most existing results establish these properties under the assumption that the game is time-independent. Recently, [Feng et al., 2023] studied the last-iterate behaviors of optimistic and extra-gradient methods in games with a time-varying payoff matrix, and proved that in an *unconstrained* periodic game, extra-gradient method converges to the equilibrium while optimistic method diverges. This finding challenges the conventional wisdom that these two methods are expected to behave similarly as they do in time-independent games. However, compared to unconstrained games, games with constrains are more common both in practical and theoretical studies. In this paper, we investigate the last-iterate behaviors of optimistic and extra-gradient methods in the *constrained* periodic games, demonstrating that similar separation results for last-iterate convergence also hold in this setting.

## 1 INTRODUCTION

Learning from repeated plays in zero-sum games has been a central research problem in game theory since the work of [Brown, 1951] and [Robinson, 1951], soon after the appearance of the minimax theorem of von Neumann. In classic normal form zero-sum games, one has to compute probability distributions $\mathbf{x}_1^* \in \Delta_n$ and $\mathbf{x}_2^* \in \Delta_m$ that consist an

equilibrium of the following problem

$$\max_{\mathbf{x}_1 \in \Delta_m} \min_{\mathbf{x}_2 \in \Delta_n} \mathbf{x}_1^\top A \mathbf{x}_2 \qquad \text{(Zero-Sum Game)}$$

where $A$ is an $n \times m$ payoff matrix, and an equilibrium $(\mathbf{x}_1^*, \mathbf{x}_2^*)$ is a pair of randomized strategies such that neither player can improve their payoff by unilaterally changing their distributions. The dynamics of online learning algorithm in games have been studied extensively. Among a variety of learning methods, Multiplicative Weights Update and Gradient Descent-Ascent, together with their optimistic and extra-gradient variants are of particular interest in *time-independent* games (i.e., the payoff matrix $A$ is time-independent).

Recently, the *last iterate property*, which captures the day-to-day behaviors of learning algorithms in games rather than their average behaviors, has attracted increasing interest due to their wide applications in machine learning and related tasks. In the regime of time-independent games, there have been quite a few results showing the last iterate convergence to Nash equilibrium in zero-sum games. Typical examples include optimistic gradient descent ascent [Daskalakis et al., 2018], extra-gradient descent ascent [Liang and Stokes, 2019] for unconstrained zero-sum games, as well as optimistic multiplicative weights update [Daskalakis and Panageas, 2018a, Fasoulakis et al., 2022], extra-gradient multiplicative weights update [Mertikopoulos et al., 2019] for constrained zero-sum games. To conclude, in the context of time-independent games, optimistic methods and extra-gradient methods exhibit similar behaviors : they both possess the last-iterate convergence property and converge by the same rate. Moreover, they can be analyzed in a unified way [Mokhtari et al., 2020].

Despite aforementioned progresses on time-independent games, only recently there have emerged researches on learning in time-varying zero-sum games [Cardoso et al., 2019, Fiez et al., 2021, Duvocelle et al., 2022, Zhang et al., 2022, Anagnostides et al., 2023, Feng et al., 2023]. In particular, it has been established by [Feng et al., 2023] that

Authors are listed according to the alphabetical order. Correspondence to: Xiao Wang <wangxiao@sufe.edu.cn>.

*Accepted for the 40th Conference on Uncertainty in Artificial Intelligence* (UAI 2024).

the optimistic gradient descent-ascent and extra gradient descent ascent have fundamentally different last iterate behaviors, unlike previous studies in time-independent zero-sum games. Nevertheless, [Feng et al., 2023] focuses on the setting of *unconstrained* zero-sum games. However, compared to unconstrained games, games with constrains are more common both in practical and theoretical studies. In this paper, we aim to address the following question:

*Is there a similar last-iterate convergence separation between optimistic and extra-gradient methods in constrained time-varying games ?*

**Our contribution.** We highlight the following two results as our main contribution :

- We construct a constrained periodic game with a common equilibrium and prove optimistic multiplicative weights update do not converge to the equilibrium in this game. See Theorem 3.1.

- We prove that if the game series in a periodic game with simplex constrains have a common equilibrium, then Extra-gradient multiplicative weights update will converge to this equilibrium. See Theorem 3.2.

By combining these two terms, we prove that there exist a clear last-iterate convergence separation between optimistic and extra-gradient methods in constrained periodic games, thereby extending the results of [Feng et al., 2023] from unconstrained to constrained settings.

**Technical Comparison.** The MWU-based algorithms considered in this paper differ from the GDA-based algorithms considered in [Feng et al., 2023] in two fundamental ways. Firstly, variations of MWU algorithms are naturally defined on the simplex constraints, allowing our analysis to avoid the difficulty of projecting onto simplex. Secondly, the algorithms considered in [Feng et al., 2023] have linear structure, i.e., can be directly analyzed as linear systems, while the MWU-based algorithms have non-linear, making the techniques of [Feng et al., 2023] ineffective in our scenario. At a high level, by considering variations of MWU algorithms, we transform the technical difficulties arising from constraints into difficulties related to analyzing non-linear dynamics of MWU-based algorithms. It is worth noting that a similar transformation can be observed in the line of research on establishing last-iterate convergence in static games: [Daskalakis et al., 2017] first proved convergence of Optimistic GDA without constraints and then [Daskalakis and Panageas, 2018a] extended their results to constrained settings for Optimistic MWU.

**Organization.** In Section 2, we present the necessary background for this work. In Section 3, we state our main results. In Section 4, we explain the main ideas behind the proof of our theoretical results. In Section 5, we provide numerical experiments to support our theoretical findings.

Discussions on possible extensions of the current results are presented in Section 6.

# 2 PRELIMINARIES

## 2.1 GAME THEORY

**Zero-sum game.** A two players zero-sum game consists of two agents $\mathcal{N} = \{1, 2\}$, and the losses of both players are determined by a payoff matrix $A \in \mathbb{R}^{m \times n}$. At each time $t$, the player 1 selects a mixed strategy $\mathbf{x}_1^t$ from the simplex constrains

$$\Delta_m = \left\{ \mathbf{x} \in \mathbb{R}^m | \sum_{i=1}^m \mathbf{x}_i = 1, \mathbf{x}_i \geq 0 \right\},$$

and similarly, the player 2 selects a mixed strategy $\mathbf{x}_2^t$ from the simplex constrains $\Delta_n$. Given that player 1 selects strategy $\mathbf{x}_1 \in \Delta_m$ and player 2 selects strategy $\mathbf{x}_2 \in \Delta_n$, player 1 receives loss $u_1(x, y) = -\langle \mathbf{x}_1, A\mathbf{x}_2 \rangle$, and player 2 receives loss $u_2(x, y) = \langle \mathbf{x}_2, A^\top \mathbf{x}_1 \rangle$. Naturally, players want to minimize their loss resulting the following min-max problem:

$$\max_{\mathbf{x}_1 \in \Delta_m} \min_{\mathbf{x}_2 \in \Delta_n} \mathbf{x}_1^\top A \mathbf{x}_2 \qquad \text{(Zero-Sum Game)}$$

A mixed strategy $\mathbf{x} \in \Delta_m$ is called fully mixed if $\mathbf{x}_i > 0$ for all $i \in [m]$. The KL-divergence between two pairs of fully mixed strategies $(\mathbf{x}, \mathbf{y})$ and $(\mathbf{x}', \mathbf{y}') \in \Delta_m \times \Delta_n$ is defined as

$$\mathrm{KL}\left((\mathbf{x}, \mathbf{y}), (\mathbf{x}', \mathbf{y}')\right) = \sum_{i \in [m]} \mathbf{x}_i \ln\left(\frac{\mathbf{x}_i}{\mathbf{x}_i'}\right) + \sum_{j \in [n]} \mathbf{y}_j \ln\left(\frac{\mathbf{y}_j}{\mathbf{y}_j'}\right).$$

The KL-divergence can be considered as a measurement of the distance between two pairs of mixed strategies. Note that for fixed $(\mathbf{x}, \mathbf{y})$, the KL-divergence will diverge to infinity when $(\mathbf{x}', \mathbf{y}')$ approaches the boundary of the simplex constrains, i,e., when some components of $(\mathbf{x}', \mathbf{y}')$ tends to zero.

**Periodic zero-sum game.** In this paper, we study games in which the payoff matrices vary over time periodically.

**Definition 2.1** (Periodic zero-sum games). *A periodic game with period $\mathcal{T}$ is an infinite sequence of zero-sum bilinear games $\{A_t\}_{t=0}^{\infty} \subset \mathbb{R}^{n \times m}$, and $A_{t+\mathcal{T}} = A_t$ for all $t \geq 1$.*

Note that the *time-independent game* is a special case of the periodic game with $\mathcal{T} = 1$. The periodic game defined here is the same as [Feng et al., 2023], except they consider the unconstrained case while we consider the constrained case. A continuous-time counterpart of the periodic zero-sum game was also studied in [Fiez et al., 2021].

## 2.2 LEARNING DYNAMICS IN GAMES

In this paper we consider two types of learning dynamics : Optimistic Multiplicative Weights Updates (OMWU)

and Extra-gradient Multiplicative Weights Updates (Extra-MWU), which are variants of the Multiplicative Weights Updates algorithms (MWU). Both (OMWU) and (Extra-MWU) possess the last-iterate convergence property in repeated game with a time-independent payoff matrix and simplex constrains, as demonstrated in previous literature [Daskalakis and Panageas, 2018a, Mertikopoulos et al., 2019, Wei et al., 2021, Fasoulakis et al., 2022]. Here we state their forms within a time-varying context.

**MWU.** The dynamics of MWU is

$$\mathbf{x}_1^{t+1} = \left( \frac{\mathbf{x}_{1,i}^t e^{\eta(A_t \mathbf{x}_2^t)^i}}{\sum_{s=1}^m \mathbf{x}_{1,s}^t e^{\eta(A_t \mathbf{x}_2^t)^s}} \right)_{i=1}^m,$$

$$\mathbf{x}_2^{t+1} = \left( \frac{\mathbf{x}_{2,j}^t e^{-\eta(A_t^\top \mathbf{x}_1^t)^j}}{\sum_{s=1}^m \mathbf{x}_{2,s}^t e^{-\eta(A_t^\top \mathbf{x}_1^t)^s}} \right)_{j=1}^n. \quad \text{(MWU)}$$

Here $\eta$ represents the step size. (MWU) belongs to the general class of Follow-the-Regularized-Leader algorithms (FTRL), which play a central role in the online learning problems [Shalev-Shwartz et al., 2012]. It is known that when two players both use (MWU) to update their strategies in a time-independent zero-sum game, the trajectories of their strategies will not converge and may diverge to the boundary of the simplex [Bailey and Piliouras, 2018].

Recently, there are also works that study the dynamical behaviors of continuous-time partner of (MWU) and more general FTRL dynamics in periodic game. It is shown that these dynamics exhibit the Poincaré recurrence property in a periodic game [Fiez et al., 2021].

**Optimistic MWU.** The dynamics of Optimistic-MWU is

$$\mathbf{x}_1^{t+1} = \left( \frac{\mathbf{x}_{1,i}^t e^{2\eta(A_t \mathbf{x}_2^t)^i - \eta(A_{t-1} \mathbf{x}_2^{t-1})^i}}{\sum_{s=1}^m \mathbf{x}_{1,s}^t e^{2\eta(A_t \mathbf{x}_2^t)^s - \eta(A_{t-1} \mathbf{x}_2^{t-1})^s}} \right)_{i=1}^m,$$

$$\mathbf{x}_2^{t+1} = \left( \frac{\mathbf{x}_{2,j}^t e^{-2\eta(A_t^\top \mathbf{x}_1^t)^j + \eta(A_{t-1}^\top \mathbf{x}_1^{t-1})^j}}{\sum_{s=1}^m \mathbf{x}_{2,s}^t e^{-2\eta(A_t^\top \mathbf{x}_1^t)^s + \eta(A_{t-1}^\top \mathbf{x}_1^{t-1})^s}} \right)_{j=1}^n. \quad \text{(OMWU)}$$

Note that in (OMWU), $t$ and $t-1$ steps are used together to update the step at time $t+1$. We will use $(\mathbf{x}_1^0, \mathbf{x}_2^0), (\mathbf{x}_1^{-1}, \mathbf{x}_2^{-1})$ to denote the initial conditions for (OMWU).

Optimistic method was proposed in [Popov, 1980] as a variant of gradient descent ascent method in saddle-point optimization problem. The last iterate convergence property of Optimistic Gradient Descent-Ascent (OGDA) in unconstrained bilinear game with a time-independent payoff was proved in [Daskalakis et al., 2017]. Recently, there are also works analyzing the regret behaviors of OGDA under a time varying setting [Anagnostides et al., 2023]. However, the study of (OMWU) in the time-varying setting is still missing in the literature, and the current work partially fills that gap.

**Extra-gradient MWU.** In Extra-MWU dynamics with a step size of $\eta$, each iteration consists of two steps. In the first step, a half step strategies vector $(\mathbf{x}_1^{t+\frac{1}{2}}, \mathbf{x}_2^{t+\frac{1}{2}})$ is calculated based on the payoff vectors in the t-th round as follows :

$$\mathbf{x}_1^{t+\frac{1}{2}} = \left( \frac{\mathbf{x}_{1,i}^t e^{\eta(A_t \mathbf{x}_2^t)^i}}{\sum_{s=1}^m \mathbf{x}_{1,s}^t e^{\eta(A_t \mathbf{x}_2^t)^s}} \right)_{i=1}^m,$$

$$\mathbf{x}_2^{t+\frac{1}{2}} = \left( \frac{\mathbf{x}_{2,j}^t e^{-\eta(A_t^\top \mathbf{x}_1^t)^j}}{\sum_{s=1}^n \mathbf{x}_{2,s}^t e^{-\eta(A_t^\top \mathbf{x}_1^t)^s}} \right)_{j=1}^n.$$

The second step for calculating the strategies $(\mathbf{x}_1^{t+1}, \mathbf{x}_2^{t+1})$ is as follows :

$$\mathbf{x}_1^{t+1} = \left( \frac{\mathbf{x}_{1,i}^t e^{\eta(A_t \mathbf{x}_2^{t+\frac{1}{2}})^i}}{\sum_{s=1}^m \mathbf{x}_{1,s}^t e^{\eta(A_t \mathbf{x}_2^{t+\frac{1}{2}})^s}} \right)_{i=1}^m,$$

$$\mathbf{x}_2^{t+1} = \left( \frac{\mathbf{x}_{2,j}^t e^{-\eta(A_t^\top \mathbf{x}_1^{t+\frac{1}{2}})^j}}{\sum_{s=1}^n \mathbf{x}_{2,s}^t e^{-\eta(A_t^\top \mathbf{x}_1^{t+\frac{1}{2}})^s}} \right)_{j=1}^n. \quad \text{(Extra-MWU)}$$

Extra-gradient was firstly proposed in [Korpelevich, 1976] as a modification of the gradient method in saddle-point optimization problem. It is known that Extra-gradient Descent-Ascent (Extra-GDA) method converge to the equilibrium in the time-independent bilinear zero-sum game with a linear convergence rate [Liang and Stokes, 2019]. Convergence of (Extra-GDA) on convex-concave game was analyzed in [Nemirovski, 2004, Monteiro and Svaiter, 2010], and convergence guarantees for special non-convex-non-concave time-independent game of the more general Extra-gradient Mirror Descent was provided in [Mertikopoulos et al., 2019].

## 2.3 RESULTS FROM DYNAMICAL SYSTEMS

In this paper, we analyze the last-iterate behavior of learning algorithms in periodic games by modeling them as dynamical systems. The resulting systems possess two characteristics that make their analysis challenging: firstly, they are non-autonomous, i.e., the evolution of the system not only depends on its current state but also on the temporal variables; secondly, they are non-linear. In this section we introduce the necessary backgrounds on this kind of dynamical systems.

**Definition 2.2** (Periodic dynamical system). *Let $\mathcal{X}$ be a subset of $\mathbb{R}^n$. A discrete, $\mathcal{T}$-periodic dynamical system is a finite sequence $f_0, ..., f_{\mathcal{T}-1}$ of maps where $f_i : \mathcal{X} \to \mathcal{X}$ for $i = 0, ..., \mathcal{T}-1$. The sequence can be extended to a periodic infinite by defining $f_i = f_{i \bmod \mathcal{T}}$ for $i \geq \mathcal{T}$. The trajectory $\{x_n\}$ of a point $x$ is given by the n-fold composition of these p maps, i.e., $x_n = f_{n-1} \circ \cdots \circ f_1 \circ f_0(x)$.*

Periodic dynamical systems are non-autonomous. The dynamical behaviors exhibited by non-autonomous systems

can be highly intricate, and typically only results pertaining to linear systems are available [Carvalho et al., 2015]. However, the study of periodic dynamical systems can be simplified by analyzing an autonomous system derived from the underlying periodic system [Franke and Selgrade, 2003, Colonius and Kliemann, 2014]. For simplicity, we present a proposition concerning the convergence behaviors of a periodic system that is useful for our analysis.

**Proposition 2.3** ([Franke and Selgrade, 2003]). *Let $\tilde{f}_i = f_{i+\mathcal{T}-1} \circ ... \circ f_i$, for $i \in [\mathcal{T}]$. Then $\tilde{f}_i$ is a time-independent dynamical system. If for all $x \in \mathcal{X}$ and each $i \in [\mathcal{T}]$, it holds that $\lim_{n \to \infty} \tilde{f}_i^n(x) = x^*$ for some $x^* \in \mathcal{X}$, then the periodic system defined by $\{f_i\}_{i=1}^{\mathcal{T}}$ will converge to $x^*$ for arbitrary initial points $x \in \mathcal{X}$.*

The proposition above demonstrates that in order to establish the convergence of a periodic system, it suffices to demonstrate the convergence of each corresponding autonomous systems $\tilde{f}_i$.

In the following, we consider the second characteristic of the dynamical systems arising from our learning algorithms : non-linearity. In general, non-linear dynamical systems exhibit complex behaviors such as chaos [Hirsch et al., 2012], thereby rendering the understanding of their global behavior challenging. Subsequently, we present results pertaining to the local behaviors of non-linear dynamical systems $\phi$ using the technique of *linearization* [Galor, 2007].

**Definition 2.4** (Stable, Unstable, and Center eigenspaces.)**.** *Let $\phi : \mathbb{R}^n \to \mathbb{R}^n$ be a continuous differentiable function, and $\bar{x}$ be a fixed point of $\phi$, i.e., $\phi(x) = x$. let $\mathcal{D}\phi(\bar{x})$ be the Jacobian matrix of $\phi$ at point $\bar{x}$. The stable eigenspace of $\bar{x}$ is defined as*

$$E^s(\bar{x}) = span\{Eigenvectors\ of\ \mathcal{D}(\bar{x})\ whose\ eigenvalues$$
$$have\ modules < 1\}.$$

*Similarly, the unstable (rep. center ) eigenspace $E^u(\bar{x})$ (rep. $E^c(\bar{x})$) of $\bar{x}$ is the subspace spanned by eigenvectors of $\mathcal{D}(\bar{x})$ whose eigenvalues have modules $> 1$ (rep. $= 1$).*

**Proposition 2.5** ( [Galor, 2007]). *Let $\phi : \mathbb{R}^n \to \mathbb{R}^n$ be a continuous differentiable function, and with the concepts defined as above, we have*

$$\dim E^s(\bar{x}) + \dim E^u(\bar{x}) + \dim E^c(\bar{x}) = n.$$

Proposition 2.5 implies that any point in $\mathbb{R}^n$ can be decomposed to linear combination of the vectors belonging to the three eigenspaces defined above. These three eigenspaces provide a full characterization on the local behavior of $\phi$ near the fixed point $\bar{x}$ : if a point $x$ is close to $\bar{x}$, and lies in the stable space $E^s(\bar{x})$, it will converge to $\bar{x}$ after sufficient number of iterations of $\phi$. On the other hand, vectors in $E^u(\bar{x})$ or $E^c(\bar{x})$ will not converge to $\bar{x}$.

# 3 MAIN RESULTS

In this section we state our main results. Under the assumption of the games in a periodic game have an unique common equilibrium, we provide an example to show that (OMWU) fails to converge to the equilibrium and even can diverge to the boundary of the simplex, as stated in Theorem 3.1. Conversely, (Extra-MWU) can converge to the equilibrium, as shown in Theorem 3.2. This distinction provides a separation on the last-iterate convergence behaviors of (OMWU) and (Extra-MWU).

**Theorem 3.1.** *For the periodic game defined by payoff matrices*

$$A_t = \begin{cases} \begin{bmatrix} 0 & 1 \\ 1 & 0 \end{bmatrix}, & t \text{ is odd} \\\\ \begin{bmatrix} 0 & -1 \\ -1 & 0 \end{bmatrix}, & t \text{ is even} \end{cases} \tag{1}$$

*and sufficient small step size $\eta$, (OMWU) has following properties :*

- *For an arbitrary small neighbourhood $\mathcal{U}$ of the equilibrium $(\boldsymbol{x}_1^*, \boldsymbol{x}_2^*)$, there exists an initial condition within $\mathcal{U}$ that causes (OMWU) to fail in converging to $(\boldsymbol{x}_1^*, \boldsymbol{x}_2^*)$.*

- *If the initial condition $(\boldsymbol{x}_1^0, \boldsymbol{x}_2^0), (\boldsymbol{x}_1^{-1}, \boldsymbol{x}_2^{-1}) \neq (\boldsymbol{x}_1^*, \boldsymbol{x}_2^*)$, then*

$$\lim_{n \to \infty} \text{KL} \left( (\boldsymbol{x}_1^*, \boldsymbol{x}_2^*), (\boldsymbol{x}_1^n, \boldsymbol{x}_2^n) \right) = +\infty.$$

It is known that in a time-independent zero-sum game, (OMWU) and its several variants will converge to the equilibrium of the game Daskalakis and Panageas [2018a,b]. The proofs for this kind of results are typically divided into two steps :

- Firstly, when $(\mathbf{x}_1^t, \mathbf{x}_2^t)$ are far from the equilibrium $(\mathbf{x}_1^*, \mathbf{x}_2^*)$, the KL-divergence $\text{KL}((\mathbf{x}_1^*, \mathbf{x}_2^*), (\mathbf{x}_1^t, \mathbf{x}_2^t))$ decreases at each step, until $(\mathbf{x}_1^t, \mathbf{x}_2^t)$ is sufficiently close to $(\mathbf{x}_1^*, \mathbf{x}_2^*)$.

- Secondly, there exists a sufficient small neighbourhood of $(\mathbf{x}_1^*, \mathbf{x}_2^*)$, such that every points in the neighbourhood will eventually converge to this equilibrium.

Theorem 3.1 implies both of these two reasons that lead to the last-iterate convergence of (OMWU) in time-independent games fail in the time-varying game defined by (1). Note that the second point in the theorem is stronger than the first point. However, to provide a clear comparison with (OMWU) in time-independent games, we state them individually.

---

Refer to the requirement for $\eta$ in Proposition A.7 in the Appendix.

In Figure (1), we present the evolution of the KL-divergence between equilibrium and strategies of players when using OMWU.

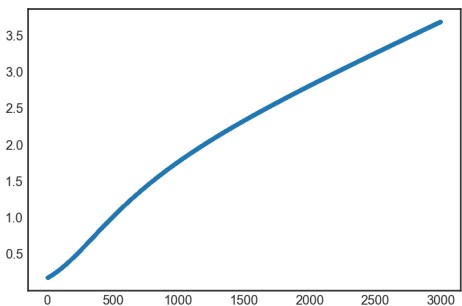

**Figure 1:** KL-divergence of OMWU in periodic game.

**Theorem 3.2.** *For a periodic game defined by the payoff matrices $\{A_t\}_{t=1}^{\mathcal{T}}$ with an unique common fully mixed equilibrium , (Extra-MWU) will converge to this equilibrium if the step size $\eta$ satisfies $\eta \cdot \max_{t\in[\mathcal{T}]}\|A_t\|< 1$.*

The last-iterate convergence property of Extra-MWU, and more generally, Extra-gradient mirror descent in time-independent game, was studied in [Mertikopoulos et al., 2019]. Note that although they referred to the algorithm they studied optimistic mirror descent, their method aligns with the Extra-gradient paradigm in the sense that the algorithm requires a two-step update in each round. The key property utilized in their proof is that the Bregman divergence (a generalization of the KL-divergence) between a fully mixed equilibrium and current strategies of players, when they use Extra-gradient mirror descent, is a decreasing function. We demonstrate that this property also holds for Extra-MWU in a periodic game if the game series in the periodic game has a common fully mixed equilibrium.

In Figure (2), we present the trajectories of strategies for a player using the Extra-MWU algorithm. The periodic game here is the same as (1). We can see that the strategy converges to the equilibrium $(0.5, 0.5)$ of the player.

# 4 OUTLINE OF THE PROOF

In this section, we outline the main steps for proving the results stated in Section 3. Further details are provided in Appendices A and B.

## 4.1 PROOFS OF THEOREM 3.1

According to the update rule of (OMWU), $\mathbf{x}_1^{t+2}$ and $\mathbf{x}_2^{t+2}$ are determined by $\mathbf{x}_1^{t+1}$, $\mathbf{x}_2^{t+1}$, $\mathbf{x}_1^t$ and $\mathbf{x}_1^t$. The mixed strategies of both players in the periodic game defined in Theorem 3.1 lie within the simplex $\Delta_2$, indicating that $\mathbf{x}_{j,2}^t$ can

---

As games with non-unique equilibrium have a measure of zero in all games, this assumption is not overly restrictive.

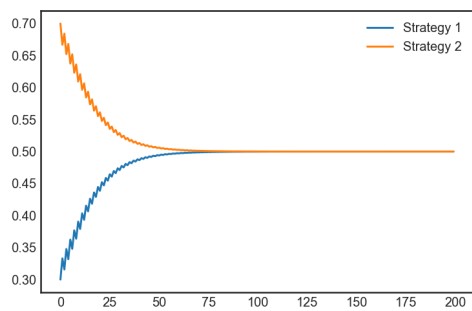

**Figure 2:** Trajectories of strategies for a player when using Extra-MWU in the periodic game defined in (1).

be determined by $\mathbf{x}_{j,1}^t$ for $j = 1, 2$ through the equation $\mathbf{x}_{j,2}^t = 1 - \mathbf{x}_{j,1}^t$. Thus, by tracing the evolution of $\mathbf{x}_{j,1}^t$, we can trace the evolution of players' mixed strategies. The dynamics of (OMWU) in Theorem 3.1 can be described equivalently by the mappings :

$$\mathcal{G}_i : (\mathbf{x}_{1,1}^t, \mathbf{x}_{1,1}^{t+1}, \mathbf{x}_{2,1}^t, \mathbf{x}_{2,1}^{t+1}) \to (\mathbf{x}_{1,1}^{t+1}, \mathbf{x}_{1,1}^{t+2}, \mathbf{x}_{2,1}^{t+1}, \mathbf{x}_{2,1}^{t+2}),$$

$i = 1, 2$, where $\mathcal{G}_1$ is the update rule for even $t$ and $\mathcal{G}_2$ is the update rule for odd $t$. Furthermore, we have

$$(\mathcal{G}_1 \circ \mathcal{G}_2)^t \left((\mathbf{x}_{1,1}^{-1}, \mathbf{x}_{1,1}^0, \mathbf{x}_{2,1}^{-1}, \mathbf{x}_{2,1}^0)\right)$$
$$= (\mathbf{x}_{1,1}^{2t-1}, \mathbf{x}_{1,1}^{2t}, \mathbf{x}_{2,1}^{2t-1}, \mathbf{x}_{2,1}^{2t}),$$

the divergence of (OMWU) can thus be deduced from the divergence of $\mathcal{G}_1 \circ \mathcal{G}_2$.

With the above construction, the proof of Theorem 3.1 is divided into three parts. Firstly, Proposition 4.1 demonstrates the existence of an initial condition in any arbitrary small neighborhood of equilibrium that does not converge to it. In the second part, Proposition 4.2 establishes that the KL-divergence monotonically increases until either $\mathbf{x}_1^t$ or $\mathbf{x}_2^t$ approaches sufficiently close to the boundary. Lastly, in the third part, Proposition 4.3 illustrates that any point close to the boundary will ultimately converge to it, which lead the KL-divergence tends to infinity.

**Proposition 4.1.** *In any arbitrary small neighbourhood $\mathcal{U}$ of the equilibrium $(\boldsymbol{x}_1^*, \boldsymbol{x}_2^*)$ of (1), there exists an initial condition in $\mathcal{U}$ such that the trajectory of (OMWU) starting from this initial condition will not converge to $(\boldsymbol{x}_1^*, \boldsymbol{x}_2^*)$.*

We prove Proposition 4.1 by calculating the eigenvalues of the Jacobi matrix of $\mathcal{G}_2 \circ \mathcal{G}_1$ at the equilibrium, which is a standard technique used in the local analysis of a dynamical system [Galor, 2007]. Similar methods are also used in proving the last-iterate convergence results for several learning algorithms in time-independent games [Daskalakis and Panageas, 2018a, Fasoulakis et al., 2022].

**Proposition 4.2.** *Under the same conditions stated in Theorem 3.1, there exists a constant $c$, which is independent of $\eta$,*

*such that for any $t \geq 3$,*

$$\text{KL}((\boldsymbol{x}_1^*, \boldsymbol{x}_2^*), (\boldsymbol{x}_1^{t+2}, \boldsymbol{x}_2^{t+2})) - \text{KL}((\boldsymbol{x}_1^*, \boldsymbol{x}_2^*), (\boldsymbol{x}_1^t, \boldsymbol{x}_2^t)) \geq c\eta^3$$

*unless either $x_1^t$ or $x_2^t$ is $\text{O}(\eta^{\frac{1}{2}})$-close to the boundary.*

We prove Proposition 4.2 by directly tracing the trajectories of mixed strategies as they evolve under (OMWU). Proposition 4.2 also implies that if the current mixed strategies used by players are far from boundary of the simplex constrains, under each iterate of (OMWU), they will steadily approach the boundary.

**Proposition 4.3.** *There exists a neighborhood $\mathcal{W}$ of the boundary of the simplex constrains such that for all $(\boldsymbol{x}_{1,1}^{-1}, \boldsymbol{x}_{1,1}^0, \boldsymbol{x}_{2,1}^{-1}, \boldsymbol{x}_{2,1}^0) \in \mathcal{W}$, we have*

$$\lim_{n \to \infty} \text{KL}((\boldsymbol{x}_{1,1}^*, \boldsymbol{x}_{1,1}^*, \boldsymbol{x}_{2,1}^*, \boldsymbol{x}_{2,1}^*),$$
$$(\mathcal{G}_1 \circ \mathcal{G}_2)^n(\boldsymbol{x}_{1,1}^{-1}, \boldsymbol{x}_{1,1}^0, \boldsymbol{x}_{2,1}^{-1}, \boldsymbol{x}_{2,1}^0)) = +\infty.$$

By combining Proposition 4.3 and Proposition 4.2, we can obtain a comprehensive understanding on the dynamics of (OMWU) in the games defined in Theorem 3.1. Firstly, when the mixed strategies are far away from the boundary of the simplex, they will rapidly approach the boundary of the simplex (Proposition 4.2). Secondly, once they are close enough to the boundary, they will be attracted to it, causing the KL-divergence tend to infinity (Proposition 4.3).

We prove Proposition 4.3 by analyzing the eigenvalues and the corresponding stable eigenspace of the Jacobian matrix of $\mathcal{G}_1 \circ \mathcal{G}_2$ at its fixed points. Interestingly, we find that these fixed points form a continuous curve, and none of the points on this curve are equilibria. This phenomenon is novel in periodic games because in time-independent games, the dynamical system modeling the learning algorithm usually only has discrete equilibrium points as fixed points [Daskalakis and Panageas, 2018a]. In Figure (3), we present these curves composed of the fixed points of $\mathcal{G}_1 \circ \mathcal{G}_2$ for different step sizes.

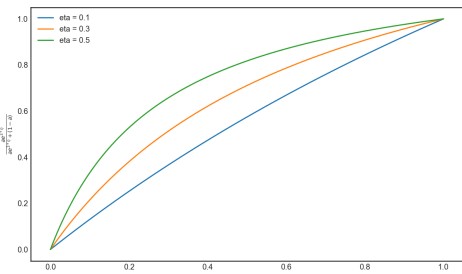

**Figure 3:** Curves composed of the fixed points of $\mathcal{G}_1 \circ \mathcal{G}_2$.

## 4.2 PROOFS OF THEOREM 3.2

Recall that in the (Extra-MWU) algorithm, each update from $(\mathbf{x}_1^t, \mathbf{x}_2^t)$ to $(\mathbf{x}_1^{t+1}, \mathbf{x}_2^{t+1})$ is divided into two steps: Firstly, an intermediate step $(\mathbf{x}_1^{t+\frac{1}{2}}, \mathbf{x}_2^{t+\frac{1}{2}})$ is calculated based on the players' payoff in the t-th round of the game. Secondly, $(\mathbf{x}_1^t, \mathbf{x}_2^t)$ and the intermediate step are used together to calculate $(\mathbf{x}_1^{t+1}, \mathbf{x}_2^{t+1})$. Since we are discussing the periodic game, the update rule of (Extra-MWU) in the current round also depends on the special payoff matrix $A_i$ for $i \in [\mathcal{T}]$ in that same round. We use

$$\mathcal{F}_i : \Delta_m \times \Delta_n \to \Delta_m \times \Delta_n$$
$$(\mathbf{x}_1^t, \mathbf{x}_2^t) \to (\mathbf{x}_1^{t+1}, \mathbf{x}_2^{t+1})$$

to denote the dynamical system determined by the (Extra-MWU) algorithm with payoff matrix $A_i$. Thus the algorithm is described by the $\mathcal{T}$-periodic dynamical system defined by $\{\mathcal{F}_i\}_{i=1}^{\mathcal{T}}$.

From Proposition 2.3, for such a periodic dynamical system, we can study its convergence property by analyzing the corresponding non-autonomous system defined as follows:

$$\tilde{\mathcal{F}}_i = \mathcal{F}_{i+\mathcal{T}-1} \circ \mathcal{F}_{i+\mathcal{T}-2} \circ ... \circ \mathcal{F}_{i+1} \circ \mathcal{F}_i,$$

where $i \in [\mathcal{T}]$. Furthermore, the periodic system converges to $(\mathbf{x}_1^*, \mathbf{x}_2^*)$ if $\tilde{\mathcal{F}}_i$ converge to $(\mathbf{x}_1^*, \mathbf{x}_2^*)$ for all $i$. Thus, the main step to prove Theorem 3.2 is to establish convergence results for $\tilde{\mathcal{F}}_i$.

For a fixed $(\mathbf{x}_1, \mathbf{x}_2)$, $\text{KL}((\mathbf{x}_1, \mathbf{x}_2), (\mathbf{x}_1', \mathbf{x}_2')) = 0$ if and only if $(\mathbf{x}_1', \mathbf{x}_2') = (\mathbf{x}_1, \mathbf{x}_2)$. Thus to prove $\tilde{\mathcal{F}}_i$ converges to the equilibrium $(\mathbf{x}_1^*, \mathbf{x}_2^*)$, it is enough to prove

$$\lim_{n \to \infty} \text{KL}((\mathbf{x}_1^*, \mathbf{x}_2^*), \mathcal{F}_i^n(\mathbf{x}_1, \mathbf{x}_2)) = 0,$$

for arbitrary initial point $(\mathbf{x}_1, \mathbf{x}_2)$. The following proposition states that in a periodic zero-sum game, the KL-divergence between the equilibrium and the current strategies decreases under an iteration of $\tilde{\mathcal{F}}_i$.

**Proposition 4.4.** *Under the same assumption as Theorem 3.2, for any $i \in [\mathcal{T}]$ and $n$, if the step size $\eta$ in (Extra-MWU) satisfies $\eta \cdot \max_{t \in [\mathcal{T}]} \|A_t\| < 1$, then we have*

$$\text{KL}\left((\boldsymbol{x}_1^*, \boldsymbol{x}_2^*), \tilde{\mathcal{F}}_i(\boldsymbol{x}_1^{n\mathcal{T}+i}, \boldsymbol{x}_2^{n\mathcal{T}+i})\right)$$
$$\leq \text{KL}\left((\boldsymbol{x}_1^*, \boldsymbol{x}_2^*), (\boldsymbol{x}_1^{n\mathcal{T}+i}, \boldsymbol{x}_2^{n\mathcal{T}+i})\right),$$

*and the equal holds if and only if $(\boldsymbol{x}_1^{n\mathcal{T}+i}, \boldsymbol{x}_2^{n\mathcal{T}+i}) = (\boldsymbol{x}_1^*, \boldsymbol{x}_2^*)$.*

The proof of Proposition 4.4 relies on a detailed analysis of the behavior of the KL-divergence under two-step method of proof of (MWU). Such a result, where the KL-divergence decreases, also plays an important role in proving convergence results for both (OMWU) and (Extra-MWU)

in static games [Mertikopoulos et al., 2019, Daskalakis and Panageas, 2018a, Fasoulakis et al., 2022].

Proposition 4.4 is not sufficient to guarantee the convergence of $\tilde{\mathcal{F}}_i$ to the equilibrium, as the rate at which the KL-divergence decreases can be slow when the current strategy is close to the equilibrium. To address this issue, we employ the following LaSalle invariance principle.

**Proposition 4.5** (LaSalle [1976])**.** *Let $G$ be any set in $\mathbb{R}^m$. Consider a difference equations system defined by a map $T : G \to G$ that is well defined for any $x \in G$ and continuous at any $x \in G$. Suppose there exists a scalar map $V : \bar{G} \to \mathbb{R}$ satisfying*

- *$V(x)$ is continuous at any $x \in \bar{G}$,*

- *$V(T(x)) - V(x) \leq 0$ for any $x \in G$.*

*For any $x_0 \in G$, if the solution to the following initial-value problem $x(n+1) = T(x(n)), x(0) = x_0$, satisfying that $\{x(n)\}_{n=1}^{\infty}$ is bounded and $x(n) \in G$ for any $n \in \mathbb{N}$, then there exists some $c \in \mathbb{R}$ such that*

$$x(n) \to M \cap V^{-1}(c)$$

*as $n \to \infty$, where $V^{-1}(c) = \{x \in \mathbb{R}^m | V(x) = c\}$, and $M$ is the largest invariant set in*

$$E = \{x \in G \mid V(T(x)) - V(x) = 0\}.$$

In our case, $\tilde{\mathcal{F}}_i$ plays the role of $T$, $\Delta_m \times \Delta_n$ plays the role of $G$, and according to Proposition 4.4, KL-divergence can serve as the scalar map $V$. The LaSalle invariance principle guarantees that the limit point under the iteration of $\tilde{\mathcal{F}}_i$ lies in the set consists of points $(\mathbf{x}_1, \mathbf{x}_2)$ that makes

$$\mathrm{KL}\left((\mathbf{x}_1^*, \mathbf{x}_2^*), \tilde{\mathcal{F}}_i(\mathbf{x}_1, \mathbf{x}_2)\right) = \mathrm{KL}\left((\mathbf{x}_1^*, \mathbf{x}_2^*), (\mathbf{x}_1, \mathbf{x}_2)\right)$$

Moreover, according to Proposition 4.4, the only possible such $(\mathbf{x}_1, \mathbf{x}_2)$ is the equilibrium point, this finish the proof that under the iteration of $\tilde{\mathcal{F}}_i$, all initial points in $\Delta_m \times \Delta_n$ will converge to the equilibrium of the periodic game. Combining this with Proposition 2.3, we can conclude that (Extra-MWU) will converge to the equilibrium.

# 5   EXPERIMENTS

In this section we provide additional numerical experiments to support our theoretical findings. In each experiments we construct periodic games with common equilibrium, and provide numerical results on the mixed strategies and KL-divegence of (Extra-MWU) and (OMWU) on these games.

## 5.1   EXPERIMENT 1

The payoff matrix in this experiment is a 2-periodic game defined by

$$A_t = \begin{cases} ((0, 0.25, 0.75), (1.5, 0, 0), (0, 1, 0)), & t \text{ is odd.} \\ \\ ((0, 0.75, 0.25), (1.5, 0, 0), (0, 0, 1)), & t \text{ is even.} \end{cases}$$

In Figure (4), we present experimental results for (Extra-MWU). In (a) of Figure (4), we can see the mixed strategy of 2-player converge to the equilibrium point $(1/3, 1/3, 1/3)$. In (b) of Figure (4), we can see $\mathrm{KL}\left((\mathbf{x}_1^*, \mathbf{x}_2^*), (\mathbf{x}_1^t, \mathbf{x}_2^t)\right) \to 0$ as time $\to \infty$. This support the result in Theorem 3.2.

In Figure (5), we present experimental results for (OMWU). In (a) of Figure (5), we can see the mixed strategy of 2-player do not converge to the equilibrium. In (b) of Figure (5), we can see $\mathrm{KL}\left((\mathbf{x}_1^*, \mathbf{x}_2^*), (\mathbf{x}_1^t, \mathbf{x}_2^t)\right) \to \infty$ as time $\to \infty$. In (b) of Figure (5), we can see $\mathrm{KL}\left((\mathbf{x}_1^*, \mathbf{x}_2^*), (\mathbf{x}_1^t, \mathbf{x}_2^t)\right) \to \infty$ as time $\to \infty$, this implies players' mixed strategies will diverge to the boundary of the simplex, thus the phenomenon here is similar to Theorem 3.1.

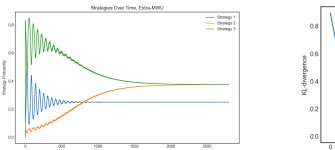 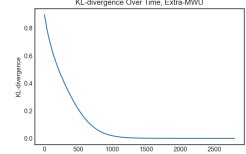

(a) Mixed strategy of 2-player  (b) KL-divergence of 2-player

**Figure 4:** First experimental results for Extra-MWU.

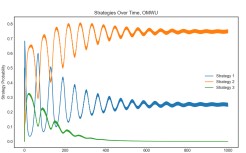 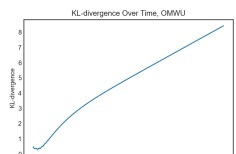

(a) Mixed strategy of 2-player  (b) KL-divergence of 2-player

**Figure 5:** First experimental results for OMWU.

## 5.2   EXPERIMENT 2

The payoff matrix in this experiment is a 4-periodic game defined by

$$A_t = \begin{cases} ((0, -1, 1), (1, 0, -1), (-1, 1, 0)), & t \bmod 4 = 0 \\ \\ ((0, 1, -1), (-1, 0, 1), (1, -1, 0)), & t \bmod 4 = 1 \\ \\ ((1, -3, 2), (-2, 1, 1), (1, 2, -3)), & t \bmod 4 = 2 \\ \\ ((1, -2, 1), (-2, 1, 1), (1, 1, -2)), & t \bmod 4 = 3 \end{cases}$$

In Figure (9), we present experimental results for (Extra-MWU). In (a) of Figure (9), we can see the mixed strategy of 2-player converge to the equilibrium point $(0.25, 0.375, 0.375)$. In (b) of Figure (9), we can see KL $((\mathbf{x}_1^*, \mathbf{x}_2^*), (\mathbf{x}_1^t, \mathbf{x}_2^t)) \to 0$ as time $\to \infty$. This support the result in Theorem 3.2.

In Figure (7), we present experimental results for (OMWU). (a) of Figure (7) shows mixed strategy do not converge to the equilibrium, and the strategy 3 tends to 0 as time process. In (b) of Figure (7), we can see KL $((\mathbf{x}_1^*, \mathbf{x}_2^*), (\mathbf{x}_1^t, \mathbf{x}_2^t)) \to \infty$ as time $\to \infty$. This implies the players mixed strategies tends to the boundary of the simplex constrains.

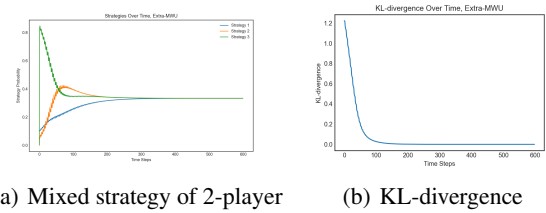

(a) Mixed strategy of 2-player     (b) KL-divergence

**Figure 6:** Second experimental results for Extra-MWU.

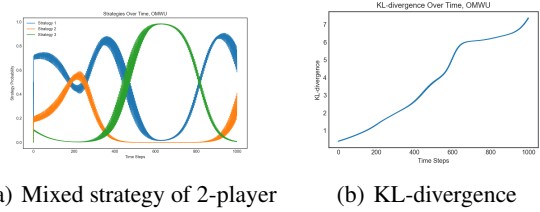

(a) Mixed strategy of 2-player     (b) KL-divergence

**Figure 7:** Second experimental results for OMWU.

# 6 GAMES WITHOUT A COMMON EQUILIBRIUM

Since the main purpose of this work is to study the last-iterate convergence behaviors of learning algorithms in the time-varying games, it is natural to require that there should be a reasonable point towards which we can hope these algorithms will converge. This is why we assume that the game series has a common equilibrium. An interesting question arises when considering what the last-iterate behaviors of (Extra-MWU) and (OMWU) look like in periodic games where there is no common equilibrium.

In the following we present several experiments to provide possible answers to this question. We summary our findings in the following :

- In a $\mathcal{T}$-periodic game without common equilibrium, (Extra-MWU) will converge to a periodic orbit with period $\mathcal{T}$. This periodic orbit will not contain equilibrium of the periodic game.

- In a $\mathcal{T}$-periodic game without common equilibrium, (OMWU) will diverge to the boundary.

In Figure (8), we present numerical results for a 3-periodic game with payoff matrices

$$
A_t = \begin{cases}
((0, -1, 1), (1, 0, -1), (-1, 1, 0)) , \ t \bmod 3 = 0 \\[2mm]
((0, 1, -1), (-1, 0, 1), (1, -1, 0)) , \ t \bmod 3 = 1 \\[2mm]
((0, 0.25, 0.75), (1.5, 0, 0), (0, 1, 0)) , \ t \bmod 3 = 2
\end{cases}
$$

For $t \bmod 3 = 0$ and $t \bmod 3 = 1$, the equilibrium is

$$
\mathbf{x}^* = \mathbf{y}^* = (1/3, 1/3, 1/3)
$$

For $t \bmod 3 = 2$, the equilibrium for $A_t$ is

$$
(\mathbf{x}^*, \mathbf{y}^*) = ((0.5, 0.25, 0.25), (0.25, 0.375, 0.375)) .
$$

In (a) of Figure 8, the three curves represent $\mathbf{x}_{1,1}^t$ when $t \bmod 3 = 0, 1$ and 2. The convergence of these three curves to distinct values suggests that the mixed strategy will converge to a periodic orbit with a period of three in (Extra-MWU). In (b) of Figure 8, it can be observed that the green curve, representing the components of the third pure strategy in the mixed strategy, converges to zero over time. This indicates that the mixed strategy tends to approach its boundary in (OMWU). These experimental findings are representative, and similar phenomena also occur in other periodic games without a common equilibrium.

We believe that the existing techniques for establishing the last-iterate convergence property are insufficient in proving that (Extra-MWU) will converge to a periodic orbit in a periodic game without a common equilibrium, as these techniques necessitate the limit state of learning algorithms being a single point [Daskalakis et al., 2017, Daskalakis and Panageas, 2018a, Mertikopoulos et al., 2019]. A possible approach to address this problem is to investigate the relationship between Extra-MWU and monotone dynamical systems, which have been shown to converge to periodic orbits in periodic environments [Hirsch and Smith, 2006].

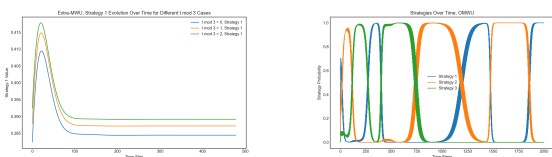

(a) Components of 1-strategy,   (b) Mixed strategy, OMWU
Extra-MWU

**Figure 8:** Experimental results for no common equilibrium.

# 7 CONCLUSION

In this paper, we investigate the last-iterate behavior of Optimistic Multiplicative Weights Updates (OMWU) and Extragradient Multiplicative Weights Updates (Extra-MWU) in periodic zero-sum games with simplex constraints. Our main findings establish a separation in the last-iterate convergence behaviors between OMWU and Extra-MWU, assuming that the game series within the periodic game. This is interesting because it challenges the conventional wisdom that these two algorithms should exhibit similar behaviors [Mokhtari et al., 2020]. Our results also extend the findings of [Feng et al., 2023] from the unconstrained setting to the more practical constrained setting. An interesting future direction is to study the dynamical behaviors of these methods in periodic games without common equilibrium.

## ACKNOWLEDGEMENTS

Ioannis Panageas would like to acknowledge startup grant from UCI and UCI ICS Research Award. Part of this work was conducted while Ioannis was visiting Archimedes Research Unit. This work has been partially supported by project MIS 5154714 of the National Recovery and Resilience Plan Greece 2.0 funded by the European Union under the NextGenerationEU Program. Xiao Wang acknowledges Grant 202110458 from Shanghai University of Finance and Economics and support from the Shanghai Research Center for Data Science and Decision Technology.

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

# Supplementary Material

**Yi Feng**[1]      **Ping Li**[1]      **Ioannis Panageas**[2]      **Xiao Wang**[1,3]

[1]Institute for Theoretical Computer Science, Shanghai University of Finance and Economics
[2]Department of Computer Science, University of California Irvine
[3]Key Laboratory of Interdisciplinary Research of Computation and Economics, Ministry of Education

## A   PROOF OF THEOREM 3.1

We introduce an equivalence of the dynamics of Optimistic MWU which tracing the first pure strategies of the two players in the case where both two players have only two pure strategies.

By the definition of the dynamics sytem of OMWU, when $t$ is even, according to the definition of $A_t$ in Theorem 3.1, we have

$$
\begin{aligned}
\mathbf{x}_{1,1}^{t+2} &= \frac{\mathbf{x}_{1,1}^{t+1} e^{2\eta(A_{t+1}\mathbf{x}_2^{t+1})^1 - \eta(A_t\mathbf{x}_2^t)^1}}{\sum_{s=1}^2 \mathbf{x}_{1,s}^{t+1} e^{2\eta(A_{t+1}\mathbf{x}_2^{t+1})^s - \eta(A_t\mathbf{x}_2^t)^s}} \\
&= \frac{\mathbf{x}_{1,1}^{t+1} e^{2\eta\mathbf{x}_{2,2}^{t+1} + \eta\mathbf{x}_{2,2}^t}}{\mathbf{x}_{1,1}^{t+1} e^{2\eta\mathbf{x}_{2,2}^{t+1} + \eta\mathbf{x}_{2,2}^t} + \mathbf{x}_{1,2}^{t+1} e^{2\eta\mathbf{x}_{2,1}^{t+1} + \eta\mathbf{x}_{2,1}^t}} \\
&= \frac{\mathbf{x}_{1,1}^{t+1} e^{2\eta(1-\mathbf{x}_{2,1}^{t+1}) + \eta(1-\mathbf{x}_{2,1}^t)}}{\mathbf{x}_{1,1}^{t+1} e^{2\eta(1-\mathbf{x}_{2,1}^{t+1}) + \eta(1-\mathbf{x}_{2,1}^t)} + (1-\mathbf{x}_{1,1}^{t+1}) e^{2\eta\mathbf{x}_{2,1}^{t+1} + \eta\mathbf{x}_{2,1}^t}} \\
&= \frac{\mathbf{x}_{1,1}^{t+1} e^{3\eta - 2\eta\mathbf{x}_{2,1}^{t+1} - \eta\mathbf{x}_{2,1}^t}}{\mathbf{x}_{1,1}^{t+1} e^{3\eta - 2\eta\mathbf{x}_{2,1}^{t+1} - \eta\mathbf{x}_{2,1}^t} + (1-\mathbf{x}_{1,1}^{t+1}) e^{2\eta\mathbf{x}_{2,1}^{t+1} + \eta\mathbf{x}_{2,1}^t}},
\end{aligned}
$$

where the third equality arises from that $\mathbf{x}_1^{t+1}$, $\mathbf{x}_1^t \in \Delta_2$. By similar computation, it holds that

$$
\mathbf{x}_{2,1}^{t+2} = \frac{\mathbf{x}_{2,1}^{t+1} e^{-3\eta + 2\eta\mathbf{x}_{1,1}^{t+1} + \eta\mathbf{x}_{1,1}^t}}{\mathbf{x}_{2,1}^{t+1} e^{-3\eta + 2\eta\mathbf{x}_{1,1}^{t+1} + \eta\mathbf{x}_{1,1}^t} + (1-\mathbf{x}_{2,1}^{t+1}) e^{-2\eta\mathbf{x}_{1,1}^{t+1} - \eta\mathbf{x}_{1,1}^t}}
$$

For the case when $t$ is odd, we have

$$
\mathbf{x}_{1,1}^{t+2} = \frac{\mathbf{x}_{1,1}^{t+1} e^{-3\eta + 2\eta\mathbf{x}_{2,1}^{t+1} + \eta\mathbf{x}_{2,1}^t}}{\mathbf{x}_{1,1}^{t+1} e^{-3\eta + 2\eta\mathbf{x}_{2,1}^{t+1} + \eta\mathbf{x}_{2,1}^t} + (1-\mathbf{x}_{1,1}^{t+1}) e^{-2\eta\mathbf{x}_{2,1}^{t+1} - \eta\mathbf{x}_{2,1}^t}},
$$

$$
\mathbf{x}_{2,1}^{t+2} = \frac{\mathbf{x}_{2,1}^{t+1} e^{3\eta - 2\eta\mathbf{x}_{1,1}^{t+1} - \eta\mathbf{x}_{1,1}^t}}{\mathbf{x}_{2,1}^{t+1} e^{3\eta - 2\eta\mathbf{x}_{1,1}^{t+1} - \eta\mathbf{x}_{1,1}^t} + (1-\mathbf{x}_{2,1}^{t+1}) e^{2\eta\mathbf{x}_{1,1}^{t+1} + \eta\mathbf{x}_{1,1}^t}}.
$$

From the above expression, it can be found that $\mathbf{x}_{1,1}^{t+2}$, $\mathbf{x}_{2,1}^{t+2}$ can be entirely determined by the vector $\{\mathbf{x}_{1,1}^t, \mathbf{x}_{1,1}^{t+1}, \mathbf{x}_{2,1}^t, \mathbf{x}_{2,1}^{t+1}\}$ in the case of when $t$ is odd and when $t$ is even.

Next, we provide two formal functions mapping $(\mathbf{x}_{1,1}^t, \mathbf{x}_{1,1}^{t+1}, \mathbf{x}_{2,1}^t, \mathbf{x}_{2,1}^{t+1})$ to $(\mathbf{x}_{1,1}^{t+1}, \mathbf{x}_{1,1}^{t+2}, \mathbf{x}_{2,1}^{t+1}, \mathbf{x}_{2,1}^{t+2})$. Let

$$
\begin{aligned}
\mathcal{G}_1 : &[0,1] \times [0,1] \times [0,1] \times [0,1] \to [0,1] \times [0,1] \times [0,1] \times [0,1] \\
&(\mathbf{z}_1, \mathbf{z}_2, \mathbf{z}_3, \mathbf{z}_4) \to \\
&\left( \mathbf{z}_2, \frac{\mathbf{z}_1 e^{3\eta - 2\eta\mathbf{z}_4 - \eta\mathbf{z}_3}}{\mathbf{z}_2 e^{3\eta - 2\eta\mathbf{z}_4 - \eta\mathbf{z}_3} + (1 - \mathbf{z}_2) e^{2\eta\mathbf{z}_4 + \eta\mathbf{z}_3}}, \right. \\
&\left. \mathbf{z}_4, \frac{\mathbf{z}_4 e^{-3\eta + 2\eta\mathbf{z}_2 + \eta\mathbf{z}_1}}{\mathbf{z}_4 e^{-3\eta + 2\eta\mathbf{z}_2 + \eta\mathbf{z}_1} + (1 - \mathbf{z}_4) e^{-2\eta\mathbf{z}_2 - \eta\mathbf{z}_1}} \right)
\end{aligned}
$$

and

$$
\begin{aligned}
\mathcal{G}_2 : &[0,1] \times [0,1] \times [0,1] \times [0,1] \to [0,1] \times [0,1] \times [0,1] \times [0,1] \\
&(\mathbf{z}_1, \mathbf{z}_2, \mathbf{z}_3, \mathbf{z}_4) \to \\
&\left( \mathbf{z}_2, \frac{\mathbf{z}_1 e^{-3\eta + 2\eta\mathbf{z}_4 + \eta\mathbf{z}_3}}{\mathbf{z}_2 e^{-3\eta + 2\eta\mathbf{z}_4 + \eta\mathbf{z}_3} + (1 - \mathbf{z}_2) e^{-2\eta\mathbf{z}_4 + \eta\mathbf{z}_3}}, \right. \\
&\left. \mathbf{z}_4, \frac{\mathbf{z}_4 e^{3\eta - 2\eta\mathbf{z}_2 - \eta\mathbf{z}_1}}{\mathbf{z}_4 e^{3\eta - 2\eta\mathbf{z}_2 - \eta\mathbf{z}_1} + (1 - \mathbf{z}_4) e^{2\eta\mathbf{z}_2 + \eta\mathbf{z}_1}} \right).
\end{aligned}
$$

According to their definition, we have

$$
\begin{aligned}
&\mathcal{G}_1 \left( (\mathbf{x}_{1,1}^t, \mathbf{x}_{1,1}^{t+1}, \mathbf{x}_{2,1}^t, \mathbf{x}_{2,1}^{t+1}) \right) \\
&= \left( \mathbf{x}_{1,1}^{t+1}, \frac{\mathbf{x}_{1,1}^{t+1} e^{3\eta - 2\eta\mathbf{x}_{2,1}^{t+1} - \eta\mathbf{x}_{2,1}^t}}{\mathbf{x}_{1,1}^{t+1} e^{3\eta - 2\eta\mathbf{x}_{2,1}^{t+1} - \eta\mathbf{x}_{2,1}^t} + (1 - \mathbf{x}_{1,1}^{t+1}) e^{2\eta\mathbf{x}_{2,1}^{t+1} + \eta\mathbf{x}_{2,1}^t}}, \right. \\
&\left. \mathbf{x}_{2,1}^{t+1}, \frac{\mathbf{x}_{2,1}^{t+1} e^{-3\eta + 2\eta\mathbf{x}_{1,1}^{t+1} + \eta\mathbf{x}_{1,1}^t}}{\mathbf{x}_{2,1}^{t+1} e^{-3\eta + 2\eta\mathbf{x}_{1,1}^{t+1} + \eta\mathbf{x}_{1,1}^t} + (1 - \mathbf{x}_{2,1}^{t+1}) e^{-2\eta\mathbf{x}_{1,1}^{t+1} - \eta\mathbf{x}_{1,1}^t}} \right).
\end{aligned}
$$

and

$$
\begin{aligned}
&\mathcal{G}_2 \left( (\mathbf{x}_{1,1}^t, \mathbf{x}_{1,1}^{t+1}, \mathbf{x}_{2,1}^t, \mathbf{x}_{2,1}^{t+1}) \right) \\
&= \left( \mathbf{x}_{1,1}^{t+1}, \frac{\mathbf{x}_{1,1}^{t+1} e^{-3\eta + 2\eta\mathbf{x}_{2,1}^{t+1} + \eta\mathbf{x}_{2,1}^t}}{\mathbf{x}_{1,1}^{t+1} e^{-3\eta + 2\eta\mathbf{x}_{2,1}^{t+1} + \eta\mathbf{x}_{2,1}^t} + (1 - \mathbf{x}_{1,1}^{t+1}) e^{-2\eta\mathbf{x}_{2,1}^{t+1} - \eta\mathbf{x}_{2,1}^t}}, \right. \\
&\left. \mathbf{x}_{2,1}^{t+1}, \frac{\mathbf{x}_{2,1}^{t+1} e^{3\eta - 2\eta\mathbf{x}_{1,1}^{t+1} - \eta\mathbf{x}_{1,1}^t}}{\mathbf{x}_{2,1}^{t+1} e^{3\eta - 2\eta\mathbf{x}_{1,1}^{t+1} - \eta\mathbf{x}_{1,1}^t} + (1 - \mathbf{x}_{2,1}^{t+1}) e^{2\eta\mathbf{x}_{1,1}^{t+1} + \eta\mathbf{x}_{1,1}^t}} \right).
\end{aligned}
$$

Combining our computation for $\mathbf{x}_{1,1}^{t+2}$ and $\mathbf{x}_{2,1}^{t+2}$ above, when $t$ is even

$$
\mathcal{G}_1 \left( (\mathbf{x}_{1,1}^t, \mathbf{x}_{1,1}^{t+1}, \mathbf{x}_{2,1}^t, \mathbf{x}_{2,1}^{t+1}) \right) = (\mathbf{x}_{1,1}^{t+1}, \mathbf{x}_{1,1}^{t+2}, \mathbf{x}_{2,1}^{t+1}, \mathbf{x}_{2,1}^{t+2})
$$

and when $t$ is odd, we have

$$
\mathcal{G}_2 \left( (\mathbf{x}_{1,1}^t, \mathbf{x}_{1,1}^{t+1}, \mathbf{x}_{2,1}^t, \mathbf{x}_{2,1}^{t+1}) \right) = (\mathbf{x}_{1,1}^{t+1}, \mathbf{x}_{1,1}^{t+2}, \mathbf{x}_{2,1}^{t+1}, \mathbf{x}_{2,1}^{t+2}).
$$

By the property of $\mathcal{G}_1$ and $\mathcal{G}_2$, it holds that

$$
(\mathbf{x}_{1,1}^{2t-1}, \mathbf{x}_{1,1}^{2t}, \mathbf{x}_{2,1}^{2t-1}, \mathbf{x}_{2,1}^{2t}) = (\mathcal{G}_1 \circ \mathcal{G}_2)^t \left( (\mathbf{x}_{1,1}^{-1}, \mathbf{x}_{1,1}^0, \mathbf{x}_{2,1}^{-1}, \mathbf{x}_{2,1}^0) \right)
$$

and

$$
(\mathbf{x}_{1,1}^{2t}, \mathbf{x}_{1,1}^{2t+1}, \mathbf{x}_{2,1}^{2t}, \mathbf{x}_{2,1}^{2t+1}) = \mathcal{G}_2 \circ (\mathcal{G}_1 \circ \mathcal{G}_2)^t \left( (\mathbf{x}_{1,1}^{-1}, \mathbf{x}_{1,1}^0, \mathbf{x}_{2,1}^{-1}, \mathbf{x}_{2,1}^0) \right).
$$

The following proposition addresses the behavior of points in the neighborhood of equilibrium for OMWU in periodic game when comparing the convergence in the OMWU in static games.

**Proposition 4.1.** *In any arbitrary small neighbourhood $\mathcal{U}$ of the equilibrium $(\boldsymbol{x}_1^*, \boldsymbol{x}_2^*)$ of (1), there exists an initial condition in $\mathcal{U}$ such that the trajectory of (OMWU) starting from this initial condition will not converge to $(\boldsymbol{x}_1^*, \boldsymbol{x}_2^*)$.*

*Proof.* By computation, the eigenvalues of the Jacobi matrix of $\mathcal{G}_1 \circ \mathcal{G}_2$ at equilibrium $(0.5, 0.5, 0.5, 0.5)$ are

- $\frac{\eta^2}{2} - \frac{\sqrt{(\eta^2+\eta+1)(\eta^2-\eta+1)}}{2} + \frac{1}{2}$,

- $\frac{\eta^2}{2} - \frac{\sqrt{(\eta^2+\eta+1)(\eta^2-\eta+1)}}{2} + \frac{1}{2}$,

- $\frac{\eta^2}{2} + \frac{\sqrt{(\eta^2+\eta+1)(\eta^2-\eta+1)}}{2} + \frac{1}{2}$,

- $\frac{\eta^2}{2} + \frac{\sqrt{(\eta^2+\eta+1)(\eta^2-\eta+1)}}{2} + \frac{1}{2}$.

It can be computed that $\frac{\eta^2}{2} + \frac{\sqrt{(\eta^2+\eta+1)(\eta^2-\eta+1)}}{2} + \frac{1}{2} > 1$ and $\frac{\eta^2}{2} - \frac{\sqrt{(\eta^2+\eta+1)(\eta^2-\eta+1)}}{2} + \frac{1}{2} < 1$. The eigenvectors in the eigenspace corresponding to $\frac{\eta^2}{2} - \frac{\sqrt{(\eta^2+\eta+1)(\eta^2-\eta+1)}}{2} + \frac{1}{2}$ can be verified to consistently have negative second elements. Thus, the decomposition of any vector in the simplex constrains must contains the eigenvectors corresponding to $\frac{\eta^2}{2} + \frac{\sqrt{(\eta^2+\eta+1)(\eta^2-\eta+1)}}{2} + \frac{1}{2}$. Therefore, by Proposition 2.5, the iterations of these vectors will diverge from the equilibrium. $\qquad\square$

Subsequently, we present compelling evidence demonstrating that the KL-divergence tends towards infinity. With the initial conditions as stated in Theorem 3.1, the proof is divided into two stages :

- **Stage 1** : we will show that when the current position is far from the boundary of $[0, 1]$, the KL-divergence will stably increase. Thus both of these variables approach to the boundary. Formally, our goal in this stage is to demonstrate the following proposition.

    **Proposition 4.2.** *Under the same conditions stated in Theorem 3.1, there exists a constant c, which is independent of $\eta$, such that for any $t \geq 3$,*

    $$\mathrm{KL}((\boldsymbol{x}_1^*, \boldsymbol{x}_2^*), (\boldsymbol{x}_1^{t+2}, \boldsymbol{x}_2^{t+2})) - \mathrm{KL}((\boldsymbol{x}_1^*, \boldsymbol{x}_2^*), (\boldsymbol{x}_1^t, \boldsymbol{x}_2^t)) \geq c\eta^3$$

    *unless either $x_1^t$ or $x_2^t$ is $\mathrm{O}(\eta^{\frac{1}{2}})$-close to the boundary.*

- **Stage 2** : The convergence of either $x_1^t$ or $x_2^t$ to the boundary results in the divergence of at least one of them towards infinity, leading to an unbounded KL-divergence. This phenomenon is described by the following proposition.

    when either $x_1^t$ or $x_2^t$ close to boundary, at least one of them will converge to the boundary, which leads to the KL-divergence goes to infinity. This can be described by the following proposition.

    **Proposition 4.3.** *There exists a neighborhood $\mathcal{W}$ of the boundary of the simplex constrains such that for all $(\boldsymbol{x}_{1,1}^{-1}, \boldsymbol{x}_{1,1}^0, \boldsymbol{x}_{2,1}^{-1}, \boldsymbol{x}_{2,1}^0) \in \mathcal{W}$, we have*

    $$\lim_{n \to \infty} \mathrm{KL}((\boldsymbol{x}_{1,1}^*, \boldsymbol{x}_{1,1}^*, \boldsymbol{x}_{2,1}^*, \boldsymbol{x}_{2,1}^*),$$
    $$(\mathcal{G}_1 \circ \mathcal{G}_2)^n (\boldsymbol{x}_{1,1}^{-1}, \boldsymbol{x}_{1,1}^0, \boldsymbol{x}_{2,1}^{-1}, \boldsymbol{x}_{2,1}^0)) = +\infty.$$

## A.1 PROOF OF STAGE 1

In this subsection, we first illustrate that under a stronger conditions than those in Proposition 4.2, the second elements of the iterative vectors $\mathbf{x}_1$ and $\mathbf{x}_2$ increase with every two time iteration. Following that, we prove that the strong condition can be generalized to the condition in Proposition 4.2. Before presenting the main lemma in this subsection, we first introduce some necessary transition expressions in the dynamic system.

**Lemma A.1.** *If $2 \mid t$, then*

1. $\frac{x_{1,1}^{t+1}}{x_{1,2}^{t+1}} = \frac{x_{1,1}^{t-1}}{x_{1,2}^{t-1}} \cdot e^{-2\eta(2x_{2,2}^t - x_{2,2}^{t-1} - x_{2,2}^{t-2})}$;

2. $\frac{x_{2,1}^{t+1}}{x_{2,2}^{t+1}} = \frac{x_{2,1}^{t-1}}{x_{2,2}^{t-1}} \cdot e^{2\eta(2x_{1,2}^t - x_{1,2}^{t-1} - x_{1,2}^{t-2})}$;

---

We utilize Matlab for computation.

3. $\dfrac{x_{1,1}^{t+2}}{x_{1,2}^{t+2}} = \dfrac{x_{1,1}^{t}}{x_{1,2}^{t}} \cdot e^{2\eta(2x_{2,2}^{t+1} - x_{2,2}^{t} - x_{2,2}^{t-1})}$;

4. $\dfrac{x_{2,1}^{t+2}}{x_{2,2}^{t+2}} = \dfrac{x_{2,1}^{t}}{x_{2,2}^{t}} \cdot e^{-2\eta(2x_{1,2}^{t+1} - x_{1,2}^{t} - x_{1,2}^{t-1})}$;

5. $\dfrac{x_{1,1}^{t+2}}{x_{1,2}^{t+2}} = \dfrac{x_{1,1}^{t+1}}{x_{1,2}^{t+1}} \cdot e^{-3\eta + 2\eta(2x_{2,2}^{t+1} + x_{2,2}^{t})}$;

6. $\dfrac{x_{2,1}^{t+1}}{x_{2,2}^{t+1}} = \dfrac{x_{2,1}^{t}}{x_{2,2}^{t}} \cdot e^{-3\eta + 2\eta(2x_{1,2}^{t} + x_{1,2}^{t-1})}$;

*Proof.* By the definition of dynamic system of OMWU, when $t$ is even,

$$
\begin{aligned}
\mathbf{x}_{1,1}^{t+1} &= \frac{\mathbf{x}_{1,1}^{t} e^{2\eta(A_t \mathbf{x}_2^t)^1 - \eta(A_{t-1}\mathbf{x}_2^{t-1})^1}}{\sum_{s=1}^{2} \mathbf{x}_{1,s}^{t} e^{2\eta(A_t \mathbf{x}_2^t)^s - \eta(A_{t-1}\mathbf{x}_2^{t-1})^s}} \\
&= \frac{\mathbf{x}_{1,1}^{t} e^{-2\eta \mathbf{x}_{2,2}^{t} - \eta \mathbf{x}_{2,2}^{t-1}}}{\mathbf{x}_{1,1}^{t} e^{-2\eta \mathbf{x}_{2,2}^{t} - \eta \mathbf{x}_{2,2}^{t-1}} + \mathbf{x}_{1,2}^{t} e^{-2\eta \mathbf{x}_{2,1}^{t} - \eta \mathbf{x}_{2,1}^{t-1}}}
\end{aligned}
\tag{2}
$$

and

$$
\begin{aligned}
\mathbf{x}_{1,2}^{t+1} &= \frac{\mathbf{x}_{1,2}^{t} e^{2\eta(A_t \mathbf{x}_2^t)^2 - \eta(A_{t-1}\mathbf{x}_2^{t-1})^2}}{\sum_{s=1}^{2} \mathbf{x}_{1,s}^{t} e^{2\eta(A_t \mathbf{x}_2^t)^s - \eta(A_{t-1}\mathbf{x}_2^{t-1})^s}} \\
&= \frac{\mathbf{x}_{1,2}^{t} e^{-2\eta \mathbf{x}_{2,1}^{t} - \eta \mathbf{x}_{2,1}^{t-1}}}{\mathbf{x}_{1,1}^{t} e^{-2\eta \mathbf{x}_{2,2}^{t} - \eta \mathbf{x}_{2,2}^{t-1}} + \mathbf{x}_{1,2}^{t} e^{-2\eta \mathbf{x}_{2,1}^{t} - \eta \mathbf{x}_{2,1}^{t-1}}} \\
&= \frac{\mathbf{x}_{1,2}^{t} e^{-3\eta + 2\eta \mathbf{x}_{2,2}^{t} + \eta \mathbf{x}_{2,2}^{t-1}}}{\mathbf{x}_{1,1}^{t} e^{-2\eta \mathbf{x}_{2,2}^{t} - \eta \mathbf{x}_{2,2}^{t-1}} + \mathbf{x}_{1,2}^{t} e^{-2\eta \mathbf{x}_{2,1}^{t} - \eta \mathbf{x}_{2,1}^{t-1}}}.
\end{aligned}
\tag{3}
$$

By the similar computation, we have

$$\mathbf{x}_{1,1}^{t} = \frac{\mathbf{x}_{1,1}^{t-1} e^{2\eta \mathbf{x}_{2,2}^{t-1} + \eta \mathbf{x}_{2,2}^{t-2}}}{\mathbf{x}_{1,1}^{t-1} e^{2\eta \mathbf{x}_{2,2}^{t-1} + \eta \mathbf{x}_{2,2}^{t-2}} + \mathbf{x}_{1,2}^{t-1} e^{3\eta - 2\eta \mathbf{x}_{2,2}^{t-1} - \eta \mathbf{x}_{2,2}^{t-2}}}, \tag{4}$$

$$\mathbf{x}_{1,2}^{t} = \frac{\mathbf{x}_{1,2}^{t-1} e^{3\eta - 2\eta \mathbf{x}_{2,2}^{t-1} - \eta \mathbf{x}_{2,2}^{t-2}}}{\mathbf{x}_{1,1}^{t-1} e^{2\eta \mathbf{x}_{2,2}^{t-1} + \eta \mathbf{x}_{2,2}^{t-2}} + \mathbf{x}_{1,2}^{t-1} e^{3\eta - 2\eta \mathbf{x}_{2,2}^{t-1} - \eta \mathbf{x}_{2,2}^{t-2}}}, \tag{5}$$

$$\mathbf{x}_{1,1}^{t+2} = \frac{\mathbf{x}_{1,1}^{t+1} e^{2\eta \mathbf{x}_{2,2}^{t+1} + \eta \mathbf{x}_{2,2}^{t}}}{\mathbf{x}_{1,1}^{t+1} e^{2\eta \mathbf{x}_{2,2}^{t+1} + \eta \mathbf{x}_{2,2}^{t}} + \mathbf{x}_{1,2}^{t+1} e^{3\eta - 2\eta \mathbf{x}_{2,2}^{t+1} - \eta \mathbf{x}_{2,2}^{t}}}, \tag{6}$$

$$\mathbf{x}_{1,2}^{t+2} = \frac{\mathbf{x}_{1,2}^{t+1} e^{3\eta - 2\eta \mathbf{x}_{2,2}^{t+1} - \eta \mathbf{x}_{2,2}^{t}}}{\mathbf{x}_{1,1}^{t+1} e^{2\eta \mathbf{x}_{2,2}^{t+1} + \eta \mathbf{x}_{2,2}^{t}} + \mathbf{x}_{1,2}^{t+1} e^{3\eta - 2\eta \mathbf{x}_{2,2}^{t+1} - \eta \mathbf{x}_{2,2}^{t}}}, \tag{7}$$

$$\mathbf{x}_{2,1}^{t} = \frac{\mathbf{x}_{2,1}^{t-1} e^{-2\eta \mathbf{x}_{1,2}^{t-1} - \eta \mathbf{x}_{1,2}^{t-2}}}{\mathbf{x}_{2,1}^{t-1} e^{-2\eta \mathbf{x}_{1,2}^{t-1} - \eta \mathbf{x}_{1,2}^{t-2}} + \mathbf{x}_{2,2}^{t-1} e^{-3\eta + 2\eta \mathbf{x}_{1,2}^{t-1} + \eta \mathbf{x}_{1,2}^{t-2}}}, \tag{8}$$

$$\mathbf{x}_{2,2}^{t} = \frac{\mathbf{x}_{2,2}^{t-1} e^{-3\eta + 2\eta \mathbf{x}_{1,2}^{t-1} + \eta \mathbf{x}_{1,2}^{t-2}}}{\mathbf{x}_{2,1}^{t-1} e^{-2\eta \mathbf{x}_{1,2}^{t-1} - \eta \mathbf{x}_{1,2}^{t-2}} + \mathbf{x}_{2,2}^{t-1} e^{-3\eta + 2\eta \mathbf{x}_{1,2}^{t-1} + \eta \mathbf{x}_{1,2}^{t-2}}}, \tag{9}$$

$$\mathbf{x}_{2,1}^{t+1} = \frac{\mathbf{x}_{2,1}^{t} e^{2\eta \mathbf{x}_{1,2}^{t} + \eta \mathbf{x}_{1,2}^{t-1}}}{\mathbf{x}_{2,1}^{t} e^{2\eta \mathbf{x}_{1,2}^{t} + \eta \mathbf{x}_{1,2}^{t-1}} + \mathbf{x}_{2,2}^{t} e^{3\eta - 2\eta \mathbf{x}_{1,2}^{t} - \eta \mathbf{x}_{1,2}^{t-1}}}, \tag{10}$$

$$\mathbf{x}_{2,2}^{t+1} = \frac{\mathbf{x}_{2,2}^{t} e^{3\eta - 2\eta \mathbf{x}_{1,2}^{t} - \eta \mathbf{x}_{1,2}^{t-1}}}{\mathbf{x}_{2,1}^{t} e^{2\eta \mathbf{x}_{1,2}^{t} + \eta \mathbf{x}_{1,2}^{t-1}} + \mathbf{x}_{2,2}^{t} e^{3\eta - 2\eta \mathbf{x}_{1,2}^{t} - \eta \mathbf{x}_{1,2}^{t-1}}}, \tag{11}$$

$$\mathbf{x}_{2,1}^{t+2} = \frac{\mathbf{x}_{2,1}^{t+1} e^{-2\eta \mathbf{x}_{1,2}^{t+1} - \eta \mathbf{x}_{1,2}^{t}}}{\mathbf{x}_{2,1}^{t+1} e^{-2\eta \mathbf{x}_{1,2}^{t+1} - \eta \mathbf{x}_{1,2}^{t}} + \mathbf{x}_{2,2}^{t+1} e^{-3\eta + 2\eta \mathbf{x}_{1,2}^{t+1} + \eta \mathbf{x}_{1,2}^{t}}}, \tag{12}$$

$$\mathbf{x}_{2,2}^{t+2} = \frac{\mathbf{x}_{2,2}^{t+1} e^{-3\eta + 2\eta \mathbf{x}_{1,2}^{t+1} + \eta \mathbf{x}_{1,2}^{t}}}{\mathbf{x}_{2,1}^{t+1} e^{-2\eta \mathbf{x}_{1,2}^{t+1} - \eta \mathbf{x}_{1,2}^{t}} + \mathbf{x}_{2,2}^{t+1} e^{-3\eta + 2\eta \mathbf{x}_{1,2}^{t+1} + \eta \mathbf{x}_{1,2}^{t}}}. \tag{13}$$

Then we have

$$\frac{\mathbf{x}_{1,1}^{t+1}}{\mathbf{x}_{1,2}^{t+1}} = \frac{\mathbf{x}_{1,1}^{t}}{\mathbf{x}_{1,2}^{t}} \cdot e^{3\eta - 2\eta(2\mathbf{x}_{2,2}^{t} + \mathbf{x}_{2,2}^{t-1})}, \tag{14}$$

$$\frac{\mathbf{x}_{1,1}^{t}}{\mathbf{x}_{1,2}^{t}} = \frac{\mathbf{x}_{1,1}^{t-1}}{\mathbf{x}_{1,2}^{t-1}} \cdot e^{-3\eta + 2\eta(2\mathbf{x}_{2,2}^{t-1} + \mathbf{x}_{2,2}^{t-2})}, \tag{15}$$

$$\frac{\mathbf{x}_{1,1}^{t+2}}{\mathbf{x}_{1,2}^{t+2}} = \frac{\mathbf{x}_{1,1}^{t+1}}{\mathbf{x}_{1,2}^{t+1}} \cdot e^{-3\eta + 2\eta(2\mathbf{x}_{2,2}^{t+1} + \mathbf{x}_{2,2}^{t})}, \tag{16}$$

$$\frac{\mathbf{x}_{2,1}^{t}}{\mathbf{x}_{2,2}^{t}} = \frac{\mathbf{x}_{2,1}^{t-1}}{\mathbf{x}_{2,2}^{t-1}} \cdot e^{3\eta - 2\eta(2\mathbf{x}_{1,2}^{t-1} + \mathbf{x}_{1,2}^{t-2})}, \tag{17}$$

$$\frac{\mathbf{x}_{2,1}^{t+1}}{\mathbf{x}_{2,2}^{t+1}} = \frac{\mathbf{x}_{2,1}^{t}}{\mathbf{x}_{2,2}^{t}} \cdot e^{-3\eta + 2\eta(2\mathbf{x}_{1,2}^{t} + \mathbf{x}_{1,2}^{t-1})}, \tag{18}$$

$$\frac{\mathbf{x}_{2,1}^{t+2}}{\mathbf{x}_{2,2}^{t+2}} = \frac{\mathbf{x}_{2,1}^{t+1}}{\mathbf{x}_{2,2}^{t+1}} \cdot e^{3\eta - 2\eta(2\mathbf{x}_{1,2}^{t+1} + \mathbf{x}_{1,2}^{t})}. \tag{19}$$

Equation (14) follows from the ratio of equation (2) to equation (3).

Equation (15) follows from the ratio of equaiton (4) to equation (5).

Equation (16) follows from the ratio of equaiton (6) to equation (7), implying item 5 in the lemma.

Equation (17) follows from the ratio of equaiton (8) to equation (9).

Equation (18) follows from the ratio of equaiton (10) to equation (11), implying item 6 in the lemma.

Equation (19) follows from the ratio of equaiton (12) to equation (13).

It holds that

$$\frac{\mathbf{x}_{1,1}^{t+1}}{\mathbf{x}_{1,2}^{t+1}} = \frac{\mathbf{x}_{1,1}^{t-1}}{\mathbf{x}_{1,2}^{t-1}} \cdot e^{-2\eta(2\mathbf{x}_{2,2}^{t} - \mathbf{x}_{2,2}^{t-1} - \mathbf{x}_{2,2}^{t-2})},$$

$$\frac{\mathbf{x}_{1,1}^{t+2}}{\mathbf{x}_{1,2}^{t+2}} = \frac{\mathbf{x}_{1,1}^{t}}{\mathbf{x}_{1,2}^{t}} \cdot e^{2\eta(2\mathbf{x}_{2,2}^{t+1} - \mathbf{x}_{2,2}^{t} - \mathbf{x}_{2,2}^{t-1})},$$

$$\frac{\mathbf{x}_{2,1}^{t+1}}{\mathbf{x}_{2,2}^{t+1}} = \frac{\mathbf{x}_{2,1}^{t-1}}{\mathbf{x}_{2,2}^{t-1}} \cdot e^{2\eta(2\mathbf{x}_{1,2}^{t} - \mathbf{x}_{1,2}^{t-1} - \mathbf{x}_{1,2}^{t-2})},$$

$$\frac{\mathbf{x}_{2,1}^{t+2}}{\mathbf{x}_{2,2}^{t+2}} = \frac{\mathbf{x}_{2,1}^{t}}{\mathbf{x}_{2,2}^{t}} \cdot e^{-2\eta(2\mathbf{x}_{1,2}^{t+1} - \mathbf{x}_{1,2}^{t} - \mathbf{x}_{1,2}^{t-1})}.$$

The first equation comes from the combination of equation 14 and equation 15, implying item 1 in the lemma.

The second equation comes from the combination of equation 14 and equation 16, implying item 3 in the lemma.

The third equation comes from the combination of equation 17 and equation 18, implying item 2 in the lemma.

The fourth equation comes from the combination of equation 18 and equation 19, implying item 4 in the lemma.

$\square$

The following lemma is a simple but useful tool in our proof.

**Lemma A.2.** *Let $u, v \in [\frac{1}{2}, 1 - \eta^{\frac{1}{2}}]$, $0 \le w \le 1$, if*

$$\frac{1-v}{v} \le \frac{1-u}{u} \cdot e^{w}, \tag{20}$$

*then we have $u - v \le w$. If*

$$\frac{1-v}{v} \ge \frac{1-u}{u} \cdot e^{w}, \tag{21}$$

*then we have $u - v \ge \frac{1}{2}w\eta^{\frac{1}{2}}$.*

*Proof.* First, we consider the case when $\frac{1-v}{v} \le \frac{1-u}{u} \cdot e^{w}$,

$$\frac{1-v}{v} \le \frac{1-u}{u} \cdot e^{w}$$
$$\le \frac{1-u}{u} \cdot (1 + 2w),$$

where the inequality comes from $e^{w} \le 1 + 2w$ when $0 < w \le 1$. This simplifies to:

$$u - v \le 2v(1-u)w$$
$$\le 2 \cdot 1 \cdot \frac{1}{2} \cdot w = w,$$

where the second line arises from $v \le 1$ and $u \ge \frac{1}{2}$.

Then we consider the second part of the lemma,

$$\frac{1-v}{v} \ge \frac{1-u}{u} \cdot e^{w}$$
$$\ge \frac{1-u}{u} \cdot (1 + w),$$

where the inequality comes from $e^w \geq 1 + w$. This simplifies to:

$$u - v \geq v(1 - u)w$$
$$\geq \frac{1}{2}w\eta^{\frac{1}{2}},$$

where the second inequality follows from $v \geq \frac{1}{2}$ and $u \leq 1 - \eta^{\frac{1}{2}}$. $\qquad\square$

The following lemma shows the effect of initial conditions on $\mathbf{x}_{1,2}^1$ and $\mathbf{x}_{2,2}^1$.

**Lemma A.3.** *For the dynamic system in Theorem 3.1, when $\mathbf{x}_{1,2}^0, \mathbf{x}_{2,2}^0 > \frac{1}{2} + 2p$ for $p \in (0, \frac{1}{4})$, and $\eta$ is sufficiently small such that $p \geq 16\eta^{\frac{1}{2}}$, then for any possible $\mathbf{x}_{1,1}^{-1}, \mathbf{x}_{1,2}^{-1}$, we have*

$$\mathbf{x}_{1,2}^1, \mathbf{x}_{2,2}^1 > \frac{1}{2} + p.$$

*Proof.* By equality (2) with $t = 0$, it holds that

$$\mathbf{x}_{1,2}^1 = \frac{\mathbf{x}_{1,2}^0 e^{-2\eta\mathbf{x}_{2,1}^0 - \eta\mathbf{x}_{2,1}^{-1}}}{\mathbf{x}_{1,1}^0 e^{-2\eta\mathbf{x}_{2,2}^0 - \eta\mathbf{x}_{2,2}^{-1}} + \mathbf{x}_{1,2}^0 e^{-2\eta\mathbf{x}_{2,1}^0 - \eta\mathbf{x}_{2,1}^{-1}}}$$

$$\geq \frac{\mathbf{x}_{1,2}^0 e^{-3\eta}}{\mathbf{x}_{1,1}^0 + \mathbf{x}_{1,2}^0}$$

$$= \mathbf{x}_{1,2}^0 e^{-3\eta} \geq (\frac{1}{2} + 2p) \cdot (1 - 3\eta) \geq \frac{1}{2} + p.$$

The last inequality comes from $p \geq 16\eta^{\frac{1}{2}}$. Applying the same method, we can also conclude that $\mathbf{x}_{2,2}^1 \geq \frac{1}{2} + p$. $\qquad\square$

Next, we give the key lemma for the proof of Proposition 4.2. Note that in comparison to Theorem 3.1, the following lemma asks a stronger condition: the initial points $\mathbf{x}_{1,2}^0$ and $\mathbf{x}_{2,2}^0$ are larger than $\frac{1}{2}$. Later, we will demonstrate how this condition can be relaxed.

**Lemma A.4.** *For the same periodic game as Theorem 3.1 defined by payoff matrices*

$$A_t = \begin{cases} \begin{bmatrix} 0 & 1 \\ 1 & 0 \end{bmatrix}, & t \text{ is odd} \\\\ \begin{bmatrix} 0 & -1 \\ -1 & 0 \end{bmatrix}, & t \text{ is even} \end{cases} \tag{22}$$

*Let $0 < p < \frac{1}{4}$, and $\eta$ sufficiently small so that $p \geq 16\eta^{\frac{1}{2}}$. And the initial points satisfy the condition*

$$\mathbf{x}_{1,2}^0, \mathbf{x}_{2,2}^0 \geq \frac{1}{2} + 2p.$$

*For $t$ is even and $t \geq 4$, if for any $k < t$, $\mathbf{x}_{1,2}^k, \mathbf{x}_{2,2}^k \leq 1 - \sqrt{\eta}$, it holds that*

1. $\frac{3}{4}p\eta^3 \leq \mathbf{x}_{2,2}^{t+1} - \mathbf{x}_{2,2}^{t-1} \leq 12\eta^2$,
2. $\frac{3}{2}p\eta^{\frac{3}{2}} \leq \mathbf{x}_{2,2}^t - \mathbf{x}_{2,2}^{t+1} \leq 3\eta$,
3. $\frac{3}{4}p\eta^3 \leq \mathbf{x}_{1,2}^{t+2} - \mathbf{x}_{1,2}^t \leq 12\eta^2$,
4. $\frac{3}{4}p\eta^3 \leq \mathbf{x}_{1,2}^{t+1} - \mathbf{x}_{1,2}^{t-1}$,
5. $\frac{3}{4}p\eta^3 \leq \mathbf{x}_{2,2}^{t+2} - \mathbf{x}_{2,2}^t$,
6. $\frac{3}{2}p\eta^{\frac{3}{2}} \leq \mathbf{x}_{1,2}^{t+1} - \mathbf{x}_{1,2}^{t+2} \leq 3\eta$.

*Proof.* By Lemma A.3, we obtain

$$\mathbf{x}_{1,2}^1, \mathbf{x}_{2,2}^1 > \frac{1}{2} + p. \tag{23}$$

We will do induction on $t$.

**Base case:**
In this part, our goal is to prove the following:

1. $\frac{3}{4}p\eta^3 \leq \mathbf{x}_{2,2}^5 - \mathbf{x}_{2,2}^3 \leq 12\eta^2,$
2. $\frac{3}{2}p\eta^{\frac{3}{2}} \leq \mathbf{x}_{2,2}^4 - \mathbf{x}_{2,2}^5 \leq 3\eta,$
3. $\frac{3}{4}p\eta^3 \leq \mathbf{x}_{1,2}^6 - \mathbf{x}_{1,2}^4 \leq 12\eta^2,$
4. $\frac{3}{4}p\eta^3 \leq \mathbf{x}_{1,2}^5 - \mathbf{x}_{1,2}^3,$
5. $\frac{3}{4}p\eta^3 \leq \mathbf{x}_{2,2}^6 - \mathbf{x}_{2,2}^4,$
6. $\frac{3}{2}p\eta^{\frac{3}{2}} \leq \mathbf{x}_{1,2}^5 - \mathbf{x}_{1,2}^6 \leq 3\eta.$

We will prove above in accordance with the sequence of the given order.

We will begin by providing an estimate for $\mathbf{x}_{1,2}^2$. From item 5 of lemma A.1 with $t = 0$, we obtain

$$\frac{\mathbf{x}_{1,1}^2}{\mathbf{x}_{1,2}^2} = \frac{\mathbf{x}_{1,1}^1}{\mathbf{x}_{1,2}^1} \cdot e^{-3\eta + 2\eta(2\mathbf{x}_{2,2}^1 + \mathbf{x}_{2,2}^0)}.$$

From $\mathbf{x}_{2,2}^0, \mathbf{x}_{2,2}^1 \in [\frac{1}{2} + p, 1]$, it holds that

$$\frac{\mathbf{x}_{1,1}^1}{\mathbf{x}_{1,2}^1} \cdot e^{6p\eta} \leq \frac{\mathbf{x}_{1,1}^2}{\mathbf{x}_{1,2}^2} \leq \frac{\mathbf{x}_{1,1}^1}{\mathbf{x}_{1,2}^1} \cdot e^{3\eta}.$$

Because $\mathbf{x}_{1,1}^1 + \mathbf{x}_{1,2}^1 = 1, \mathbf{x}_{1,1}^2 + \mathbf{x}_{1,2}^2 = 1$, combining with lemma A.2,

$$3p\eta^{\frac{3}{2}} \leq \mathbf{x}_{1,2}^1 - \mathbf{x}_{1,2}^2 \leq 3\eta,$$

which leads to

$$\mathbf{x}_{1,2}^2 \geq \mathbf{x}_{1,2}^1 - 3\eta \geq \frac{1}{2} + p - 3\eta \geq \frac{1}{2} + \frac{1}{2}p, \tag{24}$$

where the frist inequality follows from $\mathbf{x}_{1,2}^1 \in [\frac{1}{2} + p, 1]$, and the last inequality arises from $p \geq 16\eta^{\frac{1}{2}}$.

Then we provide the estimate for $\mathbf{x}_{2,2}^1 - \mathbf{x}_{2,2}^3$, by item 2 in Lemma A.1,

$$\frac{\mathbf{x}_{2,1}^3}{\mathbf{x}_{2,2}^3} = \frac{\mathbf{x}_{2,1}^1}{\mathbf{x}_{2,2}^1} \cdot e^{2\eta(2\mathbf{x}_{1,2}^2 - \mathbf{x}_{1,2}^1 - \mathbf{x}_{1,2}^0)}.$$

From inequality (24), we have $\mathbf{x}_{1,2}^2 \geq \frac{1}{2}$, thus together with $\mathbf{x}_{1,2}^1, \mathbf{x}_{1,2}^0 \in [\frac{1}{2} + p, 1]$,

$$-1 \leq 2\mathbf{x}_{1,2}^2 - \mathbf{x}_{1,2}^1 - \mathbf{x}_{1,2}^0 \leq 1.$$

Combining with Lemma A.2, we have

$$-2\eta \leq \mathbf{x}_{2,2}^1 - \mathbf{x}_{2,2}^3 \leq 2\eta. \tag{25}$$

Then we show the estimate for $\mathbf{x}_{2,2}^3 - \mathbf{x}_{2,2}^2$, by item 6 in Lemma A.1, we have

$$\frac{\mathbf{x}_{2,1}^3}{\mathbf{x}_{2,2}^3} = \frac{\mathbf{x}_{2,1}^2}{\mathbf{x}_{2,2}^2} \cdot e^{-3\eta + 2\eta(2\mathbf{x}_{1,2}^2 + \mathbf{x}_{1,2}^1)}.$$

From inequality (24), we have

$$\frac{\mathbf{x}_{2,1}^2}{\mathbf{x}_{2,2}^1} \cdot e^{4p\eta} \leq \frac{\mathbf{x}_{2,1}^3}{\mathbf{x}_{2,2}^3} \leq \frac{\mathbf{x}_{2,1}^2}{\mathbf{x}_{2,2}^1} \cdot e^{3\eta}.$$

According to Lemma A.2, we obtain

$$2p\eta^{\frac{3}{2}} \leq \mathbf{x}_{2,2}^3 - \mathbf{x}_{2,2}^2 \leq 3\eta. \tag{26}$$

Now we provide the estimate for $\mathbf{x}_{1,2}^4 - \mathbf{x}_{1,2}^2$. From item 3 in Lemma A.1,

$$\frac{\mathbf{x}_{1,1}^4}{\mathbf{x}_{1,2}^4} = \frac{\mathbf{x}_{1,1}^2}{\mathbf{x}_{1,2}^2} \cdot e^{2\eta(2\mathbf{x}_{2,2}^3 - \mathbf{x}_{2,2}^2 - \mathbf{x}_{2,2}^1)}.$$

From inequalities (25) and (26), it holds that

$$\frac{\mathbf{x}_{1,1}^2}{\mathbf{x}_{1,2}^2} \cdot e^{-2\eta^2} \leq \frac{\mathbf{x}_{1,1}^4}{\mathbf{x}_{1,2}^4} \leq \frac{\mathbf{x}_{1,1}^2}{\mathbf{x}_{1,2}^2} \cdot e^{8\eta^2}.$$

By Lemma A.2, we have

$$-8\eta^2 \leq \mathbf{x}_{1,2}^4 - \mathbf{x}_{1,2}^2 \leq 2\eta^2. \tag{27}$$

Next we provide the estimate of $\mathbf{x}_{1,2}^3 - \mathbf{x}_{1,2}^4$, by item 5 in Lemma A.1,

$$\frac{\mathbf{x}_{1,1}^4}{\mathbf{x}_{1,2}^4} = \frac{\mathbf{x}_{1,1}^3}{\mathbf{x}_{1,2}^3} \cdot e^{-3\eta + 2\eta(2\mathbf{x}_{2,2}^3 + \mathbf{x}_{2,2}^2)}.$$

According to the estimate of $\mathbf{x}_{2,2}^2$ and $\mathbf{x}_{2,2}^3$ in the inequalities (25) and (26), it holds that

$$\frac{\mathbf{x}_{1,1}^3}{\mathbf{x}_{1,2}^3} \cdot e^{3p\eta} \leq \frac{\mathbf{x}_{1,1}^4}{\mathbf{x}_{1,2}^4} \leq \frac{\mathbf{x}_{1,1}^3}{\mathbf{x}_{1,2}^3} \cdot e^{3\eta}.$$

Using Lemma A.2, we have

$$\frac{3}{2}p\eta^{\frac{3}{2}} \leq \mathbf{x}_{1,2}^3 - \mathbf{x}_{1,2}^4 \leq 3\eta. \tag{28}$$

*proof of item 1.*
According to item 2 lemma A.1 with $t = 4$, we have

$$\frac{\mathbf{x}_{2,1}^5}{\mathbf{x}_{2,2}^5} = \frac{\mathbf{x}_{2,1}^3}{\mathbf{x}_{2,2}^3} \cdot e^{2\eta(2\mathbf{x}_{1,2}^4 - \mathbf{x}_{1,2}^3 - \mathbf{x}_{1,2}^2)}.$$

Combining with the inequalities (27) and (28), it holds that

$$\frac{\mathbf{x}_{2,1}^5}{\mathbf{x}_{2,2}^3} \cdot e^{\frac{3}{2}p\eta^{\frac{5}{2}}} \leq \frac{\mathbf{x}_{2,1}^3}{\mathbf{x}_{2,2}^3} \leq \frac{\mathbf{x}_{2,1}^5}{\mathbf{x}_{2,2}^5} \cdot e^{12\eta^2}.$$

By lemma A.2, we have

$$\frac{3}{4}p\eta^3 \leq \mathbf{x}_{2,2}^5 - \mathbf{x}_{2,2}^3 \leq 12\eta^2. \tag{29}$$

*proof of item 2.*
From item 6 in lemma A.1 with $t = 4$,

$$\frac{\mathbf{x}_{2,1}^5}{\mathbf{x}_{2,2}^5} = \frac{\mathbf{x}_{2,1}^4}{\mathbf{x}_{2,2}^4} \cdot e^{-3\eta + 2\eta(2\mathbf{x}_{1,2}^4 + \mathbf{x}_{1,2}^3)}.$$

Using inequalities (27) and (28),

$$\frac{\mathbf{x}_{2,1}^4}{\mathbf{x}_{2,2}^4} \cdot e^{3p\eta} \le \frac{\mathbf{x}_{2,1}^5}{\mathbf{x}_{2,2}^5} \le \frac{\mathbf{x}_{2,1}^4}{\mathbf{x}_{2,2}^4} \cdot e^{3\eta}.$$

Then we can use lemma A.2, and obtain

$$\frac{3}{2}p\eta^{\frac{3}{2}} \le \mathbf{x}_{2,2}^4 - \mathbf{x}_{2,2}^5 \le 3\eta. \tag{30}$$

*proof of item 3.*
From item 3 in lemma A.1 with $t = 4$,

$$\frac{\mathbf{x}_{1,1}^6}{\mathbf{x}_{1,2}^6} = \frac{\mathbf{x}_{1,1}^4}{\mathbf{x}_{1,2}^4} \cdot e^{2\eta(2\mathbf{x}_{2,2}^5 - \mathbf{x}_{2,2}^4 - \mathbf{x}_{2,2}^3)}.$$

According to inequalities (29) and (30), we have

$$-3\eta \le -(\mathbf{x}_{2,2}^4 - \mathbf{x}_{2,2}^5) + (\mathbf{x}_{2,2}^5 - \mathbf{x}_{2,2}^3) \le -\frac{3}{2}p\eta^{\frac{3}{2}} + 12\eta^2,$$

which is equivalent to

$$-3\eta \le -(\mathbf{x}_{2,2}^4 - \mathbf{x}_{2,2}^5) + (\mathbf{x}_{2,2}^5 - \mathbf{x}_{2,2}^3) \le -\frac{3}{4}p\eta^{\frac{3}{2}},$$

where the right-side inequality follows from $p \ge 16\eta^{\frac{1}{2}}$. Then

$$\frac{\mathbf{x}_{1,1}^6}{\mathbf{x}_{1,2}^6} \cdot e^{2\eta \cdot \frac{3}{4}p\eta^{\frac{3}{2}}} \le \frac{\mathbf{x}_{1,1}^4}{\mathbf{x}_{1,2}^4} \le \frac{\mathbf{x}_{1,1}^6}{\mathbf{x}_{1,2}^6} \cdot e^{2\eta \cdot 3\eta},$$

$$\frac{\mathbf{x}_{1,1}^6}{\mathbf{x}_{1,2}^6} \cdot e^{\frac{3}{2}p\eta^{\frac{5}{2}}} \le \frac{\mathbf{x}_{1,1}^4}{\mathbf{x}_{1,2}^4} \le \frac{\mathbf{x}_{1,1}^6}{\mathbf{x}_{1,2}^6} \cdot e^{6\eta^2}.$$

By lemma A.2, it can be concluded that

$$\frac{3}{4}p\eta^3 \le \mathbf{x}_{1,2}^6 - \mathbf{x}_{1,2}^4 \le 6\eta^2 \le 12\eta^2.$$

*proof of item 4.*
By (29), (30), (25) and (26), we have

$$\mathbf{x}_{2,2}^4 - \frac{3}{2}p\eta^{\frac{3}{2}} \ge \mathbf{x}_{2,2}^5 \ge \mathbf{x}_{2,2}^3 \ge \mathbf{x}_{2,2}^2.$$

From item 1 in lemma A.1,

$$\frac{\mathbf{x}_{1,1}^5}{\mathbf{x}_{1,2}^5} = \frac{\mathbf{x}_{1,1}^3}{\mathbf{x}_{1,2}^3} \cdot e^{-2\eta(2\mathbf{x}_{2,2}^4 - \mathbf{x}_{2,2}^3 - \mathbf{x}_{2,2}^2)}.$$

Therefore, we obtain

$$\frac{\mathbf{x}_{1,1}^5}{\mathbf{x}_{1,2}^5} \cdot e^{6p\eta^{\frac{5}{2}}} \le \frac{\mathbf{x}_{1,1}^3}{\mathbf{x}_{1,2}^3}.$$

By Lemma A.2,

$$\mathbf{x}_{1,2}^5 - \mathbf{x}_{1,2}^3 \ge 3p\eta^3. \tag{31}$$

*proof of item 5.*
By (28) and (31),

$$\mathbf{x}_{1,2}^5 \ge \mathbf{x}_{1,2}^3 \ge \mathbf{x}_{1,2}^4 + \frac{3}{2}p\eta^{\frac{3}{2}}.$$

From item 4 in lemma A.1 with $t = 4$,

$$\frac{\mathbf{x}_{2,1}^6}{\mathbf{x}_{2,2}^6} = \frac{\mathbf{x}_{2,1}^4}{\mathbf{x}_{2,2}^4} \cdot e^{-2\eta(2\mathbf{x}_{1,2}^5 - \mathbf{x}_{1,2}^4 - \mathbf{x}_{1,2}^3)}.$$

Then it holds that

$$\frac{\mathbf{x}_{2,1}^6}{\mathbf{x}_{2,2}^6} \cdot e^{3p\eta^{\frac{5}{2}}} \leq \frac{\mathbf{x}_{2,1}^4}{\mathbf{x}_{2,2}^4}.$$

Combining with Lemma A.2, it holds that

$$\mathbf{x}_{2,2}^6 - \mathbf{x}_{2,2}^4 \geq \frac{3}{2}p\eta^3.$$

*proof of item 6.*
By (29), (30) and (25),

$$\mathbf{x}_{2,2}^4 \geq \mathbf{x}_{2,2}^5 \geq \mathbf{x}_{2,2}^3 \geq \frac{1}{2} + \frac{1}{2}p.$$

According to item 5 of lemma A.1,

$$\frac{\mathbf{x}_{1,1}^6}{\mathbf{x}_{1,2}^6} = \frac{\mathbf{x}_{1,1}^5}{\mathbf{x}_{1,2}^5} \cdot e^{-3\eta + 2\eta(2\mathbf{x}_{2,2}^5 + \mathbf{x}_{2,2}^4)},$$

which leads to

$$\frac{\mathbf{x}_{1,1}^5}{\mathbf{x}_{1,2}^5} \cdot e^{3p\eta} \leq \frac{\mathbf{x}_{1,1}^6}{\mathbf{x}_{1,2}^6} \leq \frac{\mathbf{x}_{1,1}^5}{\mathbf{x}_{1,2}^5} \cdot e^{3\eta}.$$

Combining with lemma A.2, we have

$$\frac{3}{2}p\eta^{\frac{3}{2}} \leq \mathbf{x}_{1,2}^5 - \mathbf{x}_{1,2}^6 \leq 3\eta.$$

**Induction step:**

Now assume that the result in the lemma holds for $t = k$,

1. $\frac{3}{4}p\eta^3 \leq \mathbf{x}_{2,2}^{k+1} - \mathbf{x}_{2,2}^{k-1} \leq 12\eta^2,$
2. $\frac{3}{2}p\eta^{\frac{3}{2}} \leq \mathbf{x}_{2,2}^k - \mathbf{x}_{2,2}^{k+1} \leq 3\eta,$
3. $\frac{3}{4}p\eta^3 \leq \mathbf{x}_{1,2}^{k+2} - \mathbf{x}_{1,2}^k \leq 12\eta^2,$
4. $\frac{3}{4}p\eta^3 \leq \mathbf{x}_{1,2}^{k+1} - \mathbf{x}_{1,2}^{k-1},$
5. $\frac{3}{4}p\eta^3 \leq \mathbf{x}_{2,2}^{k+2} - \mathbf{x}_{2,2}^k,$
6. $\frac{3}{2}p\eta^{\frac{3}{2}} \leq \mathbf{x}_{1,2}^{k+1} - \mathbf{x}_{1,2}^{k+2} \leq 3\eta.$

Then we prove the case of $t = k + 2$, just as follows,

1. $\frac{3}{4}p\eta^3 \leq \mathbf{x}_{2,2}^{k+3} - \mathbf{x}_{2,2}^{k+1} \leq 12\eta^2,$
2. $\frac{3}{2}p\eta^{\frac{3}{2}} \leq \mathbf{x}_{2,2}^{k+2} - \mathbf{x}_{2,2}^{k+3} \leq 3\eta,$
3. $\frac{3}{4}p\eta^3 \leq \mathbf{x}_{1,2}^{k+4} - \mathbf{x}_{1,2}^{k+2} \leq 12\eta^2,$
4. $\frac{3}{4}p\eta^3 \leq \mathbf{x}_{1,2}^{k+3} - \mathbf{x}_{1,2}^{k+1},$
5. $\frac{3}{4}p\eta^3 \leq \mathbf{x}_{2,2}^{k+4} - \mathbf{x}_{2,2}^{k+2},$
6. $\frac{3}{2}p\eta^{\frac{3}{2}} \leq \mathbf{x}_{1,2}^{k+3} - \mathbf{x}_{1,2}^{k+4} \leq 3\eta.$

*proof of item 1.*
According to item 2 in lemma A.1 with $t \to k+2$, we have

$$\frac{\mathbf{x}_{2,1}^{k+3}}{\mathbf{x}_{2,2}^{k+3}} = \frac{\mathbf{x}_{2,1}^{k+1}}{\mathbf{x}_{2,2}^{k+1}} \cdot e^{2\eta(2\mathbf{x}_{1,2}^{k+2} - \mathbf{x}_{1,2}^{k+1} - \mathbf{x}_{1,2}^{k})}.$$

Combining with the item 3 and 6 in induction assumption, it holds that

$$\frac{\mathbf{x}_{2,1}^{k+3}}{\mathbf{x}_{2,2}^{k+3}} \cdot e^{\frac{3}{2}p\eta^{\frac{5}{2}}} \le \frac{\mathbf{x}_{2,1}^{k+1}}{\mathbf{x}_{2,2}^{k+1}} \le \frac{\mathbf{x}_{2,1}^{k+3}}{\mathbf{x}_{2,2}^{k+3}} \cdot e^{6\eta^2}.$$

By lemma A.2, we have

$$\frac{3}{4}p\eta^3 \le \mathbf{x}_{2,2}^{k+3} - \mathbf{x}_{2,2}^{k+1} \le 6\eta^2 \le 12\eta^2. \tag{32}$$

*proof of item 2.*
From item 6 in lemma A.1 with $t \to k+2$,

$$\frac{\mathbf{x}_{2,1}^{k+3}}{\mathbf{x}_{2,2}^{k+3}} = \frac{\mathbf{x}_{2,1}^{k+2}}{\mathbf{x}_{2,2}^{k+2}} \cdot e^{-3\eta + 2\eta(2\mathbf{x}_{1,2}^{k+2} + \mathbf{x}_{1,2}^{k+1})}.$$

By induction assumption, we have $\mathbf{x}_{1,2}^{k+2} \ge \mathbf{x}_{1,2}^{k} \ge \mathbf{x}_{1,2}^{k-2} \ge \cdots \ge \mathbf{x}_{1,2}^{0} \ge \frac{1}{2} + p$, Using item 6 in the induction assumption, we have $\mathbf{x}_{1,2}^{k+1} \ge \mathbf{x}_{1,2}^{k+2} \ge \frac{1}{2} + p$, then

$$\frac{\mathbf{x}_{2,1}^{k+2}}{\mathbf{x}_{2,2}^{k+2}} \cdot e^{6p\eta} \le \frac{\mathbf{x}_{2,1}^{k+3}}{\mathbf{x}_{2,2}^{k+3}} \le \frac{\mathbf{x}_{2,1}^{k+2}}{\mathbf{x}_{2,2}^{k+2}} \cdot e^{3\eta}.$$

Then we can use lemma A.2, and obtain

$$\frac{3}{2}p\eta^{\frac{3}{2}} \le \mathbf{x}_{2,2}^{k+2} - \mathbf{x}_{2,2}^{k+3} \le 3\eta. \tag{33}$$

*proof of item 3.*
From item 3 in lemma A.1 with $t \to k+2$,

$$\frac{\mathbf{x}_{1,1}^{k+4}}{\mathbf{x}_{1,2}^{k+4}} = \frac{\mathbf{x}_{1,1}^{k+2}}{\mathbf{x}_{1,2}^{k+2}} \cdot e^{2\eta(2\mathbf{x}_{2,2}^{k+3} - \mathbf{x}_{2,2}^{k+2} - \mathbf{x}_{2,2}^{k+1})}.$$

and

$$2\mathbf{x}_{2,2}^{k+3} - \mathbf{x}_{2,2}^{k+2} - \mathbf{x}_{2,2}^{k+1} = -(\mathbf{x}_{2,2}^{k+2} - \mathbf{x}_{2,2}^{k+3}) + (\mathbf{x}_{2,2}^{k+3} - \mathbf{x}_{2,2}^{k+1}).$$

According to inequalities (32) and (33), we have

$$-3\eta \le -(\mathbf{x}_{2,2}^{k+2} - \mathbf{x}_{2,2}^{k+3}) + (\mathbf{x}_{2,2}^{k+3} - \mathbf{x}_{2,2}^{k+1}) \le -\frac{3}{2}p\eta^{\frac{3}{2}} + 12\eta^2,$$

which is equivalent to

$$-3\eta \le -(\mathbf{x}_{2,2}^{k+2} - \mathbf{x}_{2,2}^{k+3}) + (\mathbf{x}_{2,2}^{k+3} - \mathbf{x}_{2,2}^{k+1}) \le -\frac{3}{4}p\eta^{\frac{3}{2}}.$$

The right side inequality follows from $p \ge 16\eta^{\frac{1}{2}}$. Then

$$\frac{\mathbf{x}_{1,1}^{k+4}}{\mathbf{x}_{1,2}^{k+4}} \cdot e^{2\eta \cdot \frac{3}{4}p\eta^{\frac{3}{2}}} \le \frac{\mathbf{x}_{1,1}^{k+2}}{\mathbf{x}_{1,2}^{k+2}} \le \frac{\mathbf{x}_{1,1}^{k+4}}{\mathbf{x}_{1,2}^{k+4}} \cdot e^{2\eta \cdot 3\eta},$$

$$\frac{\mathbf{x}_{1,1}^{k+4}}{\mathbf{x}_{1,2}^{k+4}} \cdot e^{\frac{3}{2}p\eta^{\frac{5}{2}}} \le \frac{\mathbf{x}_{1,1}^{k+2}}{\mathbf{x}_{1,2}^{k+2}} \le \frac{\mathbf{x}_{1,1}^{k+4}}{\mathbf{x}_{1,2}^{k+4}} \cdot e^{6\eta^2}.$$

It can be concluded that

$$\frac{3}{4}p\eta^3 \le \mathbf{x}_{1,2}^{k+4} - \mathbf{x}_{1,2}^{k+2} \le 6\eta^2 \le 12\eta^2.$$

*proof of item 4.*
According to item 2 and item 5 in induction assumption, we have $\mathbf{x}_{2,2}^{k+2} \ge \mathbf{x}_{2,2}^{k} \ge \mathbf{x}_{2,2}^{k+1} + \frac{3}{2}p\eta^{\frac{3}{2}}$. From item 1 in lemma A.1,

$$\frac{\mathbf{x}_{1,1}^{k+3}}{\mathbf{x}_{1,2}^{k+3}} = \frac{\mathbf{x}_{1,1}^{k+1}}{\mathbf{x}_{1,2}^{k+1}} \cdot e^{-2\eta(2\mathbf{x}_{2,2}^{k+2} - \mathbf{x}_{2,2}^{k+1} - \mathbf{x}_{2,2}^{k})}$$

Therefore, we obtain

$$\frac{\mathbf{x}_{1,1}^{k+3}}{\mathbf{x}_{1,2}^{k+3}} \cdot e^{3p\eta^{\frac{5}{2}}} \le \frac{\mathbf{x}_{1,1}^{k+1}}{\mathbf{x}_{1,2}^{k+1}}.$$

Using Lemma A.2,

$$\mathbf{x}_{1,2}^{k+3} - \mathbf{x}_{1,2}^{k+1} \ge \frac{3}{2}p\eta^3. \tag{34}$$

*proof of item 5.*
By item 6 in induction assumption and (34),

$$\mathbf{x}_{1,2}^{k+3} \ge \mathbf{x}_{1,2}^{k+1} \ge \mathbf{x}_{1,2}^{k+2} + \frac{3}{2}p\eta^{\frac{3}{2}}.$$

From item 4 in lemma A.1,

$$\frac{\mathbf{x}_{2,1}^{k+4}}{\mathbf{x}_{2,2}^{k+4}} = \frac{\mathbf{x}_{2,1}^{k+2}}{\mathbf{x}_{2,2}^{k+2}} \cdot e^{-2\eta(2\mathbf{x}_{1,2}^{k+3} - \mathbf{x}_{1,2}^{k+2} - \mathbf{x}_{1,2}^{k+1})}.$$

Therefore, we obtain

$$\frac{\mathbf{x}_{2,1}^{k+4}}{\mathbf{x}_{2,2}^{k+4}} \cdot e^{3p\eta^{\frac{5}{2}}} \le \frac{\mathbf{x}_{2,1}^{k+2}}{\mathbf{x}_{2,2}^{k+2}}.$$

Then it holds that

$$\mathbf{x}_{2,2}^{k+4} - \mathbf{x}_{2,2}^{k+2} \ge \frac{3}{2}p\eta^3.$$

*proof of item 6.*
By (32), (33) and induction assumption,

$$\mathbf{x}_{2,2}^{k+2} \ge \mathbf{x}_{2,2}^{k+3} \ge \mathbf{x}_{2,2}^{k+1} \ge \frac{1}{2} + p.$$

According to item 5 in lemma A.1,

$$\frac{\mathbf{x}_{1,1}^{k+4}}{\mathbf{x}_{1,2}^{k+4}} = \frac{\mathbf{x}_{1,1}^{k+3}}{\mathbf{x}_{1,2}^{k+3}} \cdot e^{-3\eta + 2\eta(2\mathbf{x}_{2,2}^{k+3} + \mathbf{x}_{2,2}^{k+2})},$$

which leads to

$$\frac{\mathbf{x}_{1,1}^{k+3}}{\mathbf{x}_{1,2}^{k+3}} \cdot e^{6p\eta} \le \frac{\mathbf{x}_{1,1}^{k+4}}{\mathbf{x}_{1,2}^{k+4}} \le \frac{\mathbf{x}_{1,1}^{k+3}}{\mathbf{x}_{1,2}^{k+3}} \cdot e^{3\eta}.$$

Combining with lemma A.2,

$$\frac{3}{2}p\eta^{\frac{3}{2}} \le 3p\eta^{\frac{3}{2}} \le \mathbf{x}_{1,2}^{k+3} - \mathbf{x}_{1,2}^{k+4} \le 3\eta.$$

$\square$

It is natural that the closer the points are to the boundary, the larger the KL-divergence. The following lemma quantifies the KL-divergence of the points to the equilibrium and the distance between the points and the boundary in $\ell_1$ norm.

**Lemma A.5.** *For two point $x = (x_1, x_2) \in \Delta_2$, $x' = (x'_1, x'_2) \in \Delta_2$, and $c \geq 0$ such that given $p \in (0, \frac{1}{2})$*

$$x_2 - c \geq x'_2 \geq \frac{1}{2} + p,$$

*then we have*

$$\mathrm{KL}((\frac{1}{2}, \frac{1}{2}), (x_1, x_2)) - \mathrm{KL}((\frac{1}{2}, \frac{1}{2}), (x'_1, x'_2)) \geq pc.$$

*Proof.* By the definition of KL-divergence, it holds that

$$\mathrm{KL}\left((\frac{1}{2}, \frac{1}{2}), (\mathbf{x}_1, \mathbf{x}_2)\right) = \frac{1}{2}\ln(\frac{1}{2\mathbf{x}_1}) + \frac{1}{2}\ln(\frac{1}{2\mathbf{x}_2}).$$

Thus,

$$
\begin{aligned}
&\mathrm{KL}\left((\frac{1}{2}, \frac{1}{2}), (\mathbf{x}_1, \mathbf{x}_2)\right) - \mathrm{KL}\left((\frac{1}{2}, \frac{1}{2}), (\mathbf{x}'_1, \mathbf{x}'_2)\right) \\
&= \frac{1}{2}\ln\left(\frac{\mathbf{x}'_1\mathbf{x}'_2}{\mathbf{x}_1\mathbf{x}_2}\right) = \frac{1}{2}\ln\left(\frac{(1-\mathbf{x}'_2)\mathbf{x}'_2}{(1-\mathbf{x}_2)\mathbf{x}_2}\right) \\
&\geq \frac{1}{2}\ln\left(\frac{(1-\mathbf{x}_2+c)(\mathbf{x}_2-c)}{(1-\mathbf{x}_2)\mathbf{x}_2}\right) \\
&= \frac{1}{2}\ln\left(1 + \frac{2\mathbf{x}_2 - 1 - c}{(1-\mathbf{x}_2)\mathbf{x}_2}\right) \\
&\geq \frac{1}{2}\ln\left(1 + \frac{2\mathbf{x}_2 - 1 - (\mathbf{x}_2 - \frac{1}{2})}{(1-\mathbf{x}_2)\mathbf{x}_2}\right) \\
&\geq \frac{1}{2}\ln(1 + 4pc) \geq pc.
\end{aligned}
$$

The second equality arises from $\mathbf{x} = (\mathbf{x}_1, \mathbf{x}_2) \in \Delta_2$ and $\mathbf{x}' = (\mathbf{x}'_1, \mathbf{x}'_2) \in \Delta_2$. The third line follows from the decreasing monotonicity with respect to $\mathbf{x}'_2$. The fifth line and the last line are a result of $\mathbf{x}_2 - c \geq \frac{1}{2} + p$. □

When we apply Lemma A.5 to Lemma A.4, the following proposition is derived.

**Proposition A.6.** *For the periodic game defined in 1, let $p \in (0, \frac{1}{4})$ and sufficient small $\eta$ such that $p \geq 16\eta^{\frac{1}{2}}$, when $t \geq 3$, if*

$$x^0_{1,2}, x^0_{2,2} \geq \frac{1}{2} + 2p,$$

*and for any $k < t$, $x^k_{1,2}, x^k_{2,2} \leq 1 - \sqrt{\eta}$, then it holds that*

$$\mathrm{KL}\left((x^*_1, x^*_2), (x^{t+2}_1, x^{t+2}_2)\right) - \mathrm{KL}\left((x^*_1, x^*_2), (x^t_{i,1}, x^t_{i,2})\right) \geq \frac{3}{4}p^2\eta^3.$$

*Proof.* Under the conditions in the lemma, according to item 1, 3, 4, and 5 in Lemma A.4, we have for any $t \geq 3$, and $i = 1, 2$

$$\mathbf{x}^{t+2}_{i,2} - \mathbf{x}^t_{i,2} \geq \frac{3}{8}p\eta^3.$$

Futhermore, when $t$ is even,

$$\mathbf{x}^{t+2}_{i,2} > \mathbf{x}^t_{i,2} > \cdots > \mathbf{x}^4_{i,2} > \frac{1}{2} + \frac{1}{2}p,$$

where the last inequality comes from the proof in Lemma A.4 and the condition $p \geq 16\eta^{\frac{1}{2}}$. And when $t$ is odd,

$$\mathbf{x}_{i,2}^{t+2} > \mathbf{x}_{i,2}^t > \cdots > \mathbf{x}_{i,2}^3 > \frac{1}{2} + \frac{1}{2}p.$$

Then using Lemma A.5, we have that

$$\mathrm{KL}\left((\mathbf{x}_1^*, \mathbf{x}_2^*), (\mathbf{x}_1^{t+2}, \mathbf{x}_2^{t+2})\right) - \mathrm{KL}\left((\mathbf{x}_1^*, \mathbf{x}_2^*), (\mathbf{x}_1^t, \mathbf{x}_2^t)\right) \geq \frac{3}{8}p^2\eta^3.$$

$\square$

Proposition A.6 auctually states that when $\mathbf{x}_{1,2}^0, \mathbf{x}_{1,2}^0 > \frac{1}{2}$, and the iterative points never enter the neighborhood of boundary, it holds that KL-divergence will always increase with an constant depending on $\eta$ and the distance of $(\mathbf{x}_1^0, \mathbf{x}_2^0)$ to the equilibrium. Note that we have no constraint on the value of $(\mathbf{x}_1^{-1}, \mathbf{x}_2^{-1})$.

In fact, the condition $\mathbf{x}_{1,2}^0, \mathbf{x}_{1,2}^0 > \frac{1}{2}$ is not necessary. All possible cases of $(\mathbf{x}_1^0, \mathbf{x}_2^0)$ are following

1. $\mathbf{x}_{1,2}^0 > \frac{1}{2}, \mathbf{x}_{2,2}^0 > \frac{1}{2}$,
2. $\mathbf{x}_{1,2}^0 < \frac{1}{2}, \mathbf{x}_{2,2}^0 > \frac{1}{2}$,
3. $\mathbf{x}_{1,2}^0 < \frac{1}{2}, \mathbf{x}_{2,2}^0 < \frac{1}{2}$,
4. $\mathbf{x}_{1,2}^0 > \frac{1}{2}, \mathbf{x}_{2,2}^0 < \frac{1}{2}$,

Proposition A.6 states the result that under the initial condition in the first case. The key lemma in the proof of Proposition A.6 is Lemma A.4. Then combining the relation between KL-divergence and the distance to the boundary in $\ell_1$ norm, the result in the proposition is concluded.

In Figure 9(d), we present the trajectories of mixed strategies of both players under the initial condition $\mathbf{x}_1^0 = \mathbf{x}_2^0 = (0.45, 0.55)$. It illustrates the findings from Lemma A.4. Similar result are observed in the other cases as in Lemma A.4, as shown in Figure 9(a), 9(b), and 9(c).

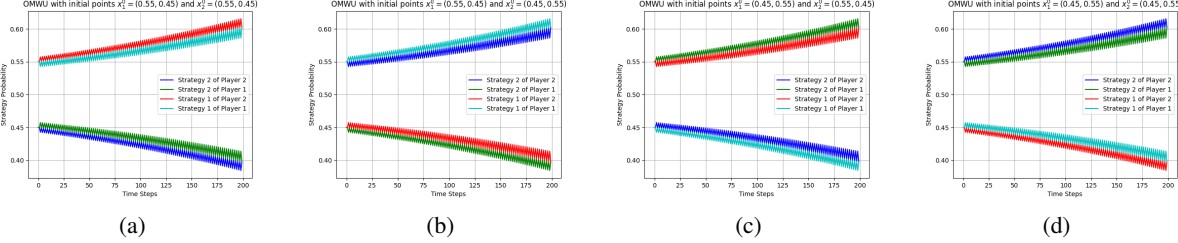

(a)          (b)          (c)          (d)

**Figure 9:** Trajectories of strategies for both players when using OMWU in the periodic game defined in (1).

Then we can extend Proposition A.6 to encompass general initial conditions:

**Proposition A.7.** *For the periodic game defined in 1, let $p = \frac{1}{2}\min(|\mathbf{x}_{1,1}^0 - \mathbf{x}_{1,1}^*|, |\mathbf{x}_{2,1}^0 - \mathbf{x}_{2,1}^*|)$, and sufficient small $\eta$ such that $p \geq 16\eta^{\frac{1}{2}}$, when $t \geq 3$, if for any $k < t$, $\mathbf{x}_{1,2}^k, \mathbf{x}_{2,2}^k \leq 1 - \sqrt{\eta}$, then it holds that*

$$\mathrm{KL}\left((\mathbf{x}_1^*, \mathbf{x}_2^*), (\mathbf{x}_1^{t+2}, \mathbf{x}_2^{t+2})\right) - \mathrm{KL}\left((\mathbf{x}_1^*, \mathbf{x}_2^*), (\mathbf{x}_1^t, \mathbf{x}_2^t)\right) \geq \frac{3}{4}p^2\eta^3.$$

Here, both $\mathbf{x}_{1,1}^*$ and $\mathbf{x}_{2,1}^*$ are equal to $\frac{1}{2}$ in game (1). Hence, parameter $p$ defined in Proposition A.7 satisfies the initial condition in Proposition A.6. The conclusions of Proposition 4.2 can be directly derived from Proposition A.7.

---

The step size of 200 is employed here to enhance the clarity in observing the increment of strategies with a probability greater than $\frac{1}{2}$ after every two iterations.

## A.2 PROOF OF STAGE 2

Lemma A.4 states that the iterative vector $(\mathbf{x}_{1,1}^n, \mathbf{x}_{1,1}^{n+1}, \mathbf{x}_{2,1}^n, \mathbf{x}_{2,1}^{n+1})^\top$ will be close to one of $(0, 0, \cdot, \cdot)$, $(1, 1, \cdot, \cdot)$, $(\cdot, \cdot, 0, 0)$, $(\cdot, \cdot, 1, 1)$ after a sufficient number of steps. Without loss of generality, let's assume that the iterative vector is close to $(0, 0, \cdot, \cdot)$. The other cases are symmetric to it.

The following lemma demonstrates that the composition of $\mathcal{G}_1 \circ \mathcal{G}_2$ has continuous fixed points on the boundary.

**Lemma A.8.** *For every* $a \in [0, 1]$, $(0, 0, a, \frac{ae^{-3\eta}}{ae^{-3\eta} + (1-a)})$ *is a fixed point of* $\mathcal{G}_1 \circ \mathcal{G}_2$.

*Proof.* Recall the definition of $\mathcal{G}_1$ and $\mathcal{G}_2$, for any $a \in (0, 1)$, we have

$$\mathcal{G}_2\left(\left(0, 0, a, \frac{ae^{-3\eta}}{ae^{-3\eta} + (1-a)}\right)\right)$$

$$= \left(0, 0, \frac{ae^{-3\eta}}{ae^{-3\eta} + (1-a)}, \frac{\frac{ae^{-3\eta}}{ae^{-3\eta}+(1-a)}e^{3\eta}}{\frac{ae^{-3\eta}}{ae^{-3\eta}+(1-a)}e^{3\eta} + (1 - \frac{ae^{-3\eta}}{ae^{-3\eta}+(1-a)})}\right)$$

$$= \left(0, 0, \frac{ae^{-3\eta}}{ae^{-3\eta} + (1-a)}, a\right),$$

and

$$\mathcal{G}_1\left(\left(0, 0, \frac{ae^{-3\eta}}{ae^{-3\eta} + (1-a)}, a\right)\right) = \left(0, 0, a, \frac{ae^{-3\eta}}{ae^{-3\eta} + (1-a)}\right).$$

Thus, it holds that

$$\mathcal{G}_1 \circ \mathcal{G}_2\left(\left(0, 0, a, \frac{ae^{-3\eta}}{ae^{-3\eta} + (1-a)}\right)\right) = \left(0, 0, a, \frac{ae^{-3\eta}}{ae^{-3\eta} + (1-a)}\right).$$

$\square$

The following lemma provides the description for the eigenvalues and eigenvectors of Jacobi matrix at the fixed points.

**Lemma A.9.** *For* $a \in (0, 1)$, *all the eigenvalues of the Jacobian matrix of* $\mathcal{G}_1 \circ \mathcal{G}_2$ *at points* $(0, 0, a, \frac{ae^{-3\eta}}{ae^{-3\eta}+(1-a)})$ *are no larger than 1. Moreover, the central eigenspace corresponds to eigenvalue 1 is generated by vectors* $(0, 0, *_1, *_2)$.

*Proof.* It can be computed that

$$0, 0, 1, e^{-\frac{2\eta a(1-a)(e^{3\eta}-1)}{a+(1-a)e^{3\eta}}}$$

are the eigenvalues of Jacobian matrix of $\mathcal{G}_1 \circ \mathcal{G}_2$ at points $(0, 0, a, \frac{ae^{-3\eta}}{ae^{-3\eta}+(1-a)})$.

Here, since $a \in (0, 1)$, it holds that $e^{-\frac{2\eta a(1-a)(e^{3\eta}-1)}{a+(1-a)e^{3\eta}}} < 1$. And the eigenvector corresponding to 1 is $\mathbf{w}_1 = (0, 0, e^{-3\eta}(a + (1-a)e^{3\eta})^2, 1)^\top$, with its first two elements being zero. $\square$

**Lemma A.10.** *Consider the composition dynamical system* $\mathcal{G}_1 \circ \mathcal{G}_2$ *which maps* $(\mathbf{x}_{1,1}^{n-1}, \mathbf{x}_{1,1}^n, \mathbf{x}_{2,1}^{n-1}, \mathbf{x}_{2,1}^n)^\top$ *to* $(\mathbf{x}_{1,1}^{n+1}, \mathbf{x}_{1,1}^{n+2}, \mathbf{x}_{2,1}^{n+1}, \mathbf{x}_{2,1}^{n+2})^\top$. *Then, for any points* $\mathbf{v}$ *close to* $(0, 0, a, \frac{ae^{-3\eta}}{ae^{-3\eta}+(1-a)})^\top$ *for arbitrary* $a \in (0, 1)$, *the first two elements of* $\mathbf{v}$ *will finally converge to zero.*

*Proof.* By Lemma A.8 for any $a \in (0, 1)$, $(0, 0, a, \frac{ae^{-3\eta}}{ae^{-3\eta}+(1-a)})$ is a fixed point of the dynamical system $\mathcal{G}_1 \circ \mathcal{G}_2$. Denote the Jacobian matrix of $\mathcal{G}_1 \circ \mathcal{G}_2$ at $(0, 0, a, \frac{ae^{-3\eta}}{ae^{-3\eta}+(1-a)})$ as $J$.

According to Lemma A.9, the neighborhood $\mathcal{W}$ of the boundary of simplex can be decomposed as follows,

$$\mathcal{W} = \mathcal{W}_1 + \mathcal{W}_0,$$

---

We employed Matlab for computation.

where $\mathcal{W}_1$ is the eigenspace corresponding to eigenvalue 1 and is spanned by $\mathbf{w}_1 = (0, 0, e^{-3\eta}(a + (1-a)e^{3\eta})^2, 1)^\top$, as mentioned in Lemma A.9. Additionally, $\mathcal{W}_0$ is the eigenspace corresponding to eigenvalues with modules smaller than 1. Naturally, for any point $\mathbf{v}$ close to $(0, 0, a, \frac{ae^{-3\eta}}{ae^{-3\eta}+(1-a)})^\top$, $\mathbf{v}$ can be decomposed as $(\mathbf{v}_1 + \mathbf{v}_0)$, where $\mathbf{v}_1 \in \mathcal{W}_1$ and $\mathbf{v}_0 \in \mathcal{W}_0$. By Proposition 2.5, with $n$ enough large, $(\mathcal{G}_1 \circ \mathcal{G}_2)^n(\mathbf{v})$ will converge to the space $(0, 0, a, \frac{ae^{-3\eta}}{ae^{-3\eta}+(1-a)})^\top + \mathcal{W}_1$. Consequently, it can be concluded that the first two elements of the iterative vector converge to zero. $\square$

With the above preparation, we are ready to prove Proposition 4.3.

*Proof of Proposition 4.3.* We only discuss about the case when $(\mathbf{x}_{1,1}^{n-1}, \mathbf{x}_{1,1}^{n}, \mathbf{x}_{2,1}^{n-1}, \mathbf{x}_{2,1}^{n})^\top$ is close to $(0, 0, \cdot, \cdot)$, while the other case are similar.

It can be computed by the update rule of dynamic of (OMWU), when $(\mathbf{x}_{1,1}^{n-1}, \mathbf{x}_{1,1}^{n}, \mathbf{x}_{2,1}^{n-1}, \mathbf{x}_{2,1}^{n})^\top$ is close to $(0, 0, \cdot, \cdot)$, it is auctually close to $(0, 0, a, \frac{ae^{-3\eta}}{ae^{-3\eta}+(1-a)})$ for some $a \in (0, 1)$.

According to Lemma A.10, when $(\mathbf{x}_{1,1}^{n-1}, \mathbf{x}_{1,1}^{n}, \mathbf{x}_{2,1}^{n-1}, \mathbf{x}_{2,1}^{n})^\top$ lies in the neighborhood of $(0, 0, a, \frac{ae^{-3\eta}}{ae^{-3\eta}+(1-a)})$, it holds that $\mathbf{x}_{1,1}^{n-1}$ and $\mathbf{x}_{1,1}^{n}$ will converge to 0. This results in the fact that KL-divergence tends to infinity. $\square$

# B  PROOF OF THEOREM 3.2

The proof is a combination of three parts:

1. The KL-divergence is a decreasing function throughout the composition of Extra-MWU in a period, starting from an arbitrary initial point. (Proposition B.12)

2. Discrete-time LaSalle invariance principle, which provides a sufficient condition for a discrete dynamical system to converge. (Proposition B.13)

3. A characterization of attractors of periodic dynamical system. (Proposition 2.3)

Let $h :\ \Delta_m \to \mathbb{R}$ be the negative entropy function, i.e., $h(\mathbf{x}) = \sum_i^m \mathbf{x}_i \ln \mathbf{x}_i$ , and $h^*(\cdot)$ be the convex conjugate of $h(\cdot)$, i.e.,

$$h^* :\ \mathbb{R}^m \to \mathbb{R}$$
$$\mathbf{y} \to \max_{\mathbf{x} \in \Delta_m} \{\langle \mathbf{y}, \mathbf{x} \rangle - h(\mathbf{x})\}.$$

Note that we have $\nabla h(\mathbf{x}) = (1 + \ln(\mathbf{x}_i))_{i=1}^m$.

**Lemma B.1** (Page 148 in Shalev-Shwartz et al. [2012]). *We have*

$$h^*(\mathbf{y}) = \ln \left( \sum_{s=1}^m e^{\mathbf{y}_s} \right), \ \nabla h^*(\mathbf{y}) = \left( \frac{e^{\mathbf{y}_i}}{\sum_{s=1}^m e^{\mathbf{y}_s}} \right)_{i=1}^m.$$

**Definition B.2.** *We define the equivalence relation " $\sim$ " between two vectors in $\mathbb{R}^m$ as follows : For two vectors $\mathbf{y}$ and $\mathbf{y}' \in \mathbb{R}^m$,*

$$\mathbf{y} \sim \mathbf{y}' \iff \exists\, \mathbf{c} = (c, ..., c) \in \mathbb{R}^m,$$
$$\text{such that } \mathbf{y} - \mathbf{y}' = \mathbf{c}.$$

*We denote the space generated by $\mathbb{R}^m$ module the above equivalence relation as $\mathbb{R}^m/ \sim$, and use $[\mathbf{y}]$ to represent the equivalence class that $\mathbf{y}$ lies in.*

**Remark B.3.** *With the equivalence relation defined above, the function $\nabla h^*$ can be thought as a function defined on $\mathbb{R}^m/ \sim$, i.e.,*

$$\nabla h^* : \mathbb{R}^m/ \sim\ \to \Delta_m$$
$$[\mathbf{y}] \to \left( \frac{e^{\mathbf{y}_i}}{\sum_{s=1}^m e^{\mathbf{y}_s}} \right)_{i=1}^m.$$

*Moreover, the function $\nabla h(\cdot)$ can be thought as a function take values in $\mathbb{R}^m / \sim$, i.e.,*

$$\nabla h : \Delta_m \to \mathbb{R}^m / \sim$$
$$\boldsymbol{x} \to [(1 + \ln(\boldsymbol{x}_i))_{i=1}^m].$$

In the following, for a vector $\mathbf{y} \in \mathbb{R}^m$, we will use $[\mathbf{y}]$ to represent to equivalence class in $\mathbb{R}^m / \sim$ that $\mathbf{y}$ lies in.

**Lemma B.4.** *$\nabla h^*(\cdot)$ and $\nabla h(\cdot)$ are inverse functions to each other, i.e., we have both $\nabla h^* \circ \nabla h : \Delta_m \to \Delta_m$ and $\nabla h \circ \nabla h^* : \mathbb{R}^m / \sim \to \mathbb{R}^m / \sim$ are identity maps.*

*Proof.* It is directly to verify

$$[\mathbf{y}] \xrightarrow{\nabla h^*} \left( \frac{e^{\mathbf{y}_i}}{\sum_{s=1}^m e^{\mathbf{y}_s}} \right) \xrightarrow{\nabla h} \left[ \left( 1 + \ln\left(\frac{e^{\mathbf{y}_i}}{\sum_{s=1}^m e^{\mathbf{y}_s}}\right) \right)_{s=1}^m \right].$$

and note that $\left[ \left( 1 + \ln\left(\frac{e^{\mathbf{y}_i}}{\sum_{s=1}^m e^{\mathbf{y}_s}}\right) \right)_{s=1}^m \right] = [\mathbf{y}]$, thus $\nabla h \circ \nabla h^* = \mathrm{Id}$.

It is also similar to verify $\nabla h^* \circ \nabla h = \mathrm{Id}$. $\qquad\square$

**Lemma B.5.** *$h : \Delta_m^\circ \to \mathbb{R}$ is 1-strongly convex and has 1-Lipschitz continuous gradients, and $h^*(\cdot)$ is 1-strongly convex and has 1-Lipschitz continuous gradients.*

*Proof.* It can be computed that for $i \in [n]$

$$\frac{\partial h}{\partial \mathbf{x}_i} = \ln(\mathbf{x}_i) + 1, \quad \frac{\partial^2 h}{\partial \mathbf{x}_i^2} = \frac{1}{\mathbf{x}_i}, \quad \frac{\partial^2 h}{\partial \mathbf{x}_i \partial \mathbf{x}_j} = 0.$$

From $\mathbf{x} \in \Delta_m^\circ$, we have that $\frac{1}{\mathbf{x}_i} \geq 1$. So $h$ is diagonal matrix with each diagonal element larger than 1. Then we have that $h$ is 1-strongly convex and $\nabla h$ is 1-Lipschitz continuous. The statemens about $h^*(\cdot)$ follows from the standard Fenchel duality property, for example, see Theorem 1 in Zhou [2018]. $\qquad\square$

The vanilla Multiplicative Weights Updates algorithm (MWU) for one player can be written as the following function :

$$\mathrm{MWU} : \Delta_m \times \mathbb{R}^m / \sim \to \Delta_m$$
$$(\mathbf{x}, [\mathbf{y}]) \to \left( \frac{\mathbf{x}_i e^{\mathbf{y}_i}}{\sum_{s=1}^m \mathbf{x}_s e^{\mathbf{y}_s}} \right)_{i=1}^m.$$

**Definition B.6.** *We define a function $\phi : \Delta_m \times \mathbb{R}^m / \sim \to \mathbb{R}^m / \sim$ as follow:*

$$\phi : \Delta_m \times \mathbb{R}^m / \sim \to \mathbb{R}^m / \sim$$
$$(\boldsymbol{x}, [\boldsymbol{y}]) \to [\nabla h(\boldsymbol{x}) + \boldsymbol{y}].$$

**Proposition B.7.** *The following diagram is commutative :*

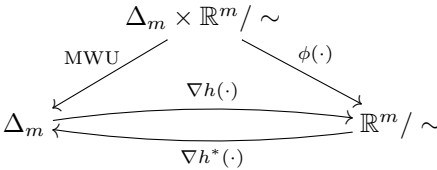

*Proof.* For any $(\mathbf{x}, [\mathbf{y}]) \in \Delta_m \times \mathbb{R}^m / \sim$, our goal is to prove that

1. $\nabla h \circ \mathrm{MWU}(\mathbf{x}, [\mathbf{y}]) = \phi(\mathbf{x}, [\mathbf{y}])$,
2. $\nabla h^* \circ \phi(\mathbf{x}, [\mathbf{y}]) = \mathrm{MWU}(\mathbf{x}, [\mathbf{y}])$.

We start by proving the first item. It is directly to calculate

$$\nabla h \circ \mathrm{MWU}(\mathbf{x}, \mathbf{y}) = \nabla h \left( \left( \frac{\mathbf{x}_i e^{\mathbf{y}_i}}{\sum_{s=1}^{m} \mathbf{x}_s e^{\mathbf{y}_s}} \right)_{i=1}^{m} \right)$$

$$= \left( 1 + \ln \left( \frac{\mathbf{x}_i e^{\mathbf{y}_i}}{\sum_{s=1}^{m} \mathbf{x}_s e^{\mathbf{y}_s}} \right) \right)_{i=1}^{m}$$

$$= \left[ \left( 1 + \mathbf{y}_i + \ln(\mathbf{x}_i) - \ln(\sum_{s=1}^{m} \mathbf{x}_s e^{\mathbf{y}_s}) \right)_{i=1}^{m} \right]$$

$$= \left[ (\mathbf{y}_i + \ln(\mathbf{x}_i))_{i=1}^{m} \right],$$

and

$$\phi(\mathbf{x}, [\mathbf{y}]) = [(1 + \ln(\mathbf{x}_i) + \mathbf{y}_i)].$$

Since as equivalence class, we have $[(\mathbf{y}_i + \ln(\mathbf{x}_i))_{i=1}^{m}] = [(1 + \ln(\mathbf{x}_i) + \mathbf{y}_i)_{i=1}^{m}]$, this prove the first item.

For the second item, it is directly to calculate

$$\nabla h^* \circ \phi(\mathbf{x}, [\mathbf{y}]) = \nabla h^* \left( [(1 + \ln(\mathbf{x}_i) + \mathbf{y}_i)_{i=1}^{m}] \right)$$

$$= \nabla h^* \left( [(\ln(\mathbf{x}_i) + \mathbf{y}_i)_{i=1}^{m}] \right)$$

$$= \left( \frac{e^{\ln(\mathbf{x}_i) + \mathbf{y}_i}}{\sum_{s=1}^{m} e^{\ln(\mathbf{x}_s) + \mathbf{y}_s}} \right)_{i=1}^{m}$$

$$= \left( \frac{\mathbf{x}_i e^{\mathbf{y}_i}}{\sum_{s=1}^{m} \mathbf{x}_s e^{\mathbf{y}_s}} \right)$$

$$= \mathrm{MWU}(\mathbf{x}, [\mathbf{y}]).$$

This prove the second item.

$\square$

**Lemma B.8.** *For arbitrary* $\boldsymbol{p} \in \Delta_m$, *if* $\boldsymbol{y} \in \mathbb{R}^m$ *and* $\boldsymbol{x} = \nabla h^*([\boldsymbol{y}])$, *then*

$$\langle \nabla h(\boldsymbol{x}) - \boldsymbol{y}, \boldsymbol{x} - \boldsymbol{p} \rangle = 0.$$

*Proof.* From Lemma B.4, we have

$$\nabla h(\nabla h^*([\mathbf{y}])) = \mathbf{y} + \mathbf{c},$$

where $\mathbf{c}$ is a constant vector, therefore $\langle \nabla h(\mathbf{x}) - \mathbf{y}, \mathbf{x} - \mathbf{p} \rangle$ can be transitioned to

$$\langle \mathbf{c}, \mathbf{x} - \mathbf{p} \rangle$$
$$= \langle \mathbf{c}, \mathbf{x} \rangle - \langle \mathbf{c}, \mathbf{p} \rangle$$
$$= 0.$$

The second equality arises from $\mathbf{x}$ and $\mathbf{p}$ belong to the simplex, i.e., $\sum \mathbf{x}_i = \sum \mathbf{p}_i = 1$. $\square$

**Lemma B.9.** *We have*

$$\| \mathrm{MWU}(\boldsymbol{x}, [\boldsymbol{y}_1]) - \mathrm{MWU}(\boldsymbol{x}, [\boldsymbol{y}_2]) \| \leq \| \boldsymbol{y}_1 - \boldsymbol{y}_2 \|.$$

*Proof.* From Proposition B.7, we have

$$\| \mathrm{MWU}(\mathbf{x}, [\mathbf{y}_1]) - \mathrm{MWU}(\mathbf{x}, [\mathbf{y}_2]) \| = \| \nabla h^* (\phi(\mathbf{x}, \mathbf{y}_1)) - \nabla h^* (\phi(\mathbf{x}, \mathbf{y}_2)) \|$$
$$\leq \| \phi(\mathbf{x}, \mathbf{y}_1) - \phi(\mathbf{x}, \mathbf{y}_2) \|$$
$$= \| \mathbf{y}_1 - \mathbf{y}_2 \|,$$

where the first inequality from $h^*$ has 1-Lipschitz continuous gradient, see Lemma B.5, and the last equality is from the definition of $\phi$. $\square$

**Lemma B.10** (Three-points identity Chen and Teboulle [1993]). *For ant $\boldsymbol{p}, \boldsymbol{x}, \boldsymbol{x}' \in \Delta_m$, the following equality holds*

$$\mathrm{KL}(\boldsymbol{p}, \boldsymbol{x}') = \mathrm{KL}(\boldsymbol{p}, \boldsymbol{x}) + \mathrm{KL}(\boldsymbol{x}, \boldsymbol{x}') + \langle (\ln(x_i'/x_i)_{i=1}^m), (x_i - p_i)_{i=1}^m \rangle$$

*Proof.* By definiton, it holds that

$$\mathrm{KL}(\mathbf{p}, \mathbf{x}') = h(\mathbf{p}) - h(\mathbf{x}') - \langle \nabla h(\mathbf{x}'), \mathbf{p} - \mathbf{x}' \rangle,$$
$$\mathrm{KL}(\mathbf{p}, \mathbf{x}) = h(\mathbf{p}) - h(\mathbf{x}) - \langle \nabla h(\mathbf{x}), \mathbf{p} - \mathbf{x} \rangle,$$
$$\mathrm{KL}(\mathbf{x}, \mathbf{x}') = h(\mathbf{x}) - h(\mathbf{x}') - \langle \nabla h(\mathbf{x}'), \mathbf{x} - \mathbf{x}' \rangle.$$

Then $\mathrm{KL}(\mathbf{x}, \mathbf{x}') + \mathrm{KL}(\mathbf{p}, \mathbf{x}) - \mathrm{KL}(\mathbf{p}, \mathbf{x}')$ gives

$$\mathrm{KL}(\mathbf{p}, \mathbf{x}') = \mathrm{KL}(\mathbf{p}, \mathbf{x}) + \mathrm{KL}(\mathbf{x}, \mathbf{x}') + \langle \nabla h(\mathbf{x}') - \nabla h(\mathbf{x}), \mathbf{p} - \mathbf{x} \rangle.$$

By replacing $\nabla h(\mathbf{x})$ with $(\ln \mathbf{x}_i + 1)_{i=1}^m$, the result can be concluded. $\qquad\square$

**Lemma B.11.** *Let $\boldsymbol{x}^\dagger = \mathrm{MWU}(\boldsymbol{x}, \boldsymbol{y})$, then*

$$\mathrm{KL}(\boldsymbol{p}, \boldsymbol{x}^\dagger) = \mathrm{KL}(\boldsymbol{p}, \boldsymbol{x}) - \mathrm{KL}(\boldsymbol{x}^\dagger, \boldsymbol{x}) + \langle \boldsymbol{y}, \boldsymbol{x}^\dagger - \boldsymbol{p} \rangle.$$

*Proof.* In Lemma B.10, take $\mathbf{x} = \mathbf{x}^\dagger$ and $\mathbf{x}' = \mathbf{x}$, it turns out to be

$$\mathrm{KL}(\mathbf{p}, \mathbf{x}^\dagger) = \mathrm{KL}(\mathbf{p}, \mathbf{x}) - \mathrm{KL}(\mathbf{x}^\dagger, \mathbf{x}) + \langle \nabla h(\mathbf{x}) - \nabla h(\mathbf{x}^\dagger), \mathbf{x}^\dagger - \mathbf{p} \rangle$$
$$= \mathrm{KL}(\mathbf{p}, \mathbf{x}) - \mathrm{KL}(\mathbf{x}^\dagger, \mathbf{x}) + \langle \nabla h(\mathbf{x}) - \phi(\mathbf{x}, [\mathbf{y}]), \mathbf{x}^\dagger - \mathbf{p} \rangle$$
$$= \mathrm{KL}(\mathbf{p}, \mathbf{x}) - \mathrm{KL}(\mathbf{x}^\dagger, \mathbf{x}) + \langle \mathbf{y}, \mathbf{x}^\dagger - \mathbf{p} \rangle,$$

where the second equality comes from Proposition B.7 and the last equality is from the definition of $\phi$ and the fact that for any two vectors $\mathbf{y}, \mathbf{y}' \in [\mathbf{y}]$, we have $\langle \mathbf{y}, \mathbf{p} \rangle = \langle \mathbf{y}', \mathbf{p} \rangle$. $\qquad\square$

Let $\mathcal{F}_i : \Delta_m \times \Delta_n \to \Delta_m \times \Delta_n$ be the (Extra-MWU) algorithm with payoff matrix $A_i$, for any initial condition $(\mathbf{x}_0, \mathbf{y}_0)$ and any $i \in [\mathcal{T}]$, the following Property shows the KL-divergence will decrease after an iteration by

$$\tilde{\mathcal{F}}_i = \mathcal{F}_{i+\mathcal{T}-1} \circ \mathcal{F}_{i+\mathcal{T}-2} \circ ... \circ \mathcal{F}_{i+1} \circ \mathcal{F}_i.$$

**Proposition B.12.** *For any $i \in [\mathcal{T}]$ and $n$, if the step size $\eta$ in (Extra-MWU) satisfies $\eta \cdot \max_{t \in [\mathcal{T}]} \|A_t\| < 1$, then we have*

$$\mathrm{KL}\left((\boldsymbol{x}_1^*, \boldsymbol{x}_2^*), \tilde{\mathcal{F}}_i(\boldsymbol{x}_1^{n\mathcal{T}+i}, \boldsymbol{x}_2^{n\mathcal{T}+i})\right) < \mathrm{KL}\left((\boldsymbol{x}_1^*, \boldsymbol{x}_2^*), (\boldsymbol{x}_1^{n\mathcal{T}+i}, \boldsymbol{x}_2^{n\mathcal{T}+i})\right),$$

*and the equal holds if and only if $(\boldsymbol{x}_1^{n\mathcal{T}+i}, \boldsymbol{x}_2^{n\mathcal{T}+i}) = (\boldsymbol{x}_1^*, \boldsymbol{x}_2^*)$.*

*Proof.* In fact, from $\tilde{\mathcal{F}}_i(\mathbf{x}_1^{n\mathcal{T}+i}, \mathbf{x}_2^{n\mathcal{T}+i}) = (\mathbf{x}_1^{(n+1)\mathcal{T}+i}, \mathbf{x}_2^{(n+1)\mathcal{T}+i})$, it holds that

$$\mathrm{KL}\left((\mathbf{x}_1^*, \mathbf{x}_2^*), \tilde{\mathcal{F}}_i(\mathbf{x}_1^{n\mathcal{T}+i}, \mathbf{x}_2^{n\mathcal{T}+i})\right) - \mathrm{KL}\left((\mathbf{x}_1^*, \mathbf{x}_2^*), (\mathbf{x}_1^{n\mathcal{T}+i}, \mathbf{x}_2^{n\mathcal{T}+i})\right)$$
$$= \mathrm{KL}\left((\mathbf{x}_1^*, \mathbf{x}_2^*), (\mathbf{x}_1^{(n+1)\mathcal{T}+i}, \mathbf{x}_2^{(n+1)\mathcal{T}+i})\right) - \mathrm{KL}\left((\mathbf{x}_1^*, \mathbf{x}_2^*), (\mathbf{x}_1^{n\mathcal{T}+i}, \mathbf{x}_2^{n\mathcal{T}+i})\right)$$
$$= \sum_{j=0}^{\mathcal{T}-1} \left( \mathrm{KL}\left((\mathbf{x}_1^*, \mathbf{x}_2^*), (\mathbf{x}_1^{n\mathcal{T}+i+j+1}, \mathbf{x}_2^{n\mathcal{T}+i+j+1})\right) - \mathrm{KL}\left((\mathbf{x}_1^*, \mathbf{x}_2^*), (\mathbf{x}_1^{n\mathcal{T}+i+j}, \mathbf{x}_2^{n\mathcal{T}+i+j})\right) \right).$$

In the following we will prove for any $j \in [\mathcal{T}]$, we have

$$\mathrm{KL}\left((\mathbf{x}_1^*, \mathbf{x}_2^*), (\mathbf{x}_1^{n\mathcal{T}+i+j+1}, \mathbf{x}_2^{n\mathcal{T}+i+j+1})\right) - \mathrm{KL}\left((\mathbf{x}_1^*, \mathbf{x}_2^*), (\mathbf{x}_1^{n\mathcal{T}+i+j}, \mathbf{x}_2^{n\mathcal{T}+i+j})\right) < 0,$$

which implies Proposition B.12.

In following, for a fixed $j \in [\mathcal{T}]$, we use $\mathbf{x}$ to represent $(\mathbf{x}_1^{n\mathcal{T}+i+j}, \mathbf{x}_2^{n\mathcal{T}+i+j})$, $\mathbf{x}^\dagger$ to represent $(\mathbf{x}_1^{n\mathcal{T}+i+j+\frac{1}{2}}, \mathbf{x}_2^{n\mathcal{T}+i+j+\frac{1}{2}})$, and $\mathbf{x}^\ddagger$ to represent $(\mathbf{x}_1^{n\mathcal{T}+i+j+1}, \mathbf{x}_2^{n\mathcal{T}+i+j+1})$. Similarly, we use $\mathbf{y}$ to represent $(\mathbf{y}_1^{n\mathcal{T}+i+j}, \mathbf{y}_2^{n\mathcal{T}+i+j})$, $\mathbf{y}^\dagger$ to represent $(\mathbf{y}_1^{n\mathcal{T}+i+j+\frac{1}{2}}, \mathbf{y}_2^{n\mathcal{T}+i+j+\frac{1}{2}})$.

By the definition of (Extra-MWU), for $i \in [2]$ we have

$$\mathbf{x}_i^{n\mathcal{T}+i+j+\frac{1}{2}} = \mathrm{MWU}(\mathbf{x}_i^{n\mathcal{T}+i+j}, \mathbf{y}_i^{n\mathcal{T}+i+j}),$$
$$\mathbf{x}_i^{n\mathcal{T}+i+j+1} = \mathrm{MWU}(\mathbf{x}_i^{n\mathcal{T}+i+j}, \mathbf{y}_i^{n\mathcal{T}+i+j+\frac{1}{2}}),$$

which leads to

$$\mathbf{x}_i^\dagger = \mathrm{MWU}(\mathbf{x}_i, \mathbf{y}_i),$$
$$\mathbf{x}_i^\ddagger = \mathrm{MWU}(\mathbf{x}_i, \mathbf{y}_i^\dagger).$$

Replacing $\mathbf{x}^\dagger$ with $\mathbf{x}^\ddagger$ and $\mathbf{p}$ with $\mathbf{x}^*$ in Lemma B.11, we have

$$\mathrm{KL}(\mathbf{x}^*, \mathbf{x}^\ddagger) - \mathrm{KL}(\mathbf{x}^*, \mathbf{x}^\dagger) = -\mathrm{KL}(\mathbf{x}^\ddagger, \mathbf{x}) + \langle \mathbf{y}^\dagger, \mathbf{x}^\ddagger - \mathbf{x}^* \rangle.$$

Let $\mathbf{p} = \mathbf{x}^\ddagger$ in Lemma B.11, we have

$$\mathrm{KL}(\mathbf{x}^\ddagger, \mathbf{x}) = \mathrm{KL}(\mathbf{x}^\ddagger, \mathbf{x}^\dagger) + \mathrm{KL}(\mathbf{x}^\dagger, \mathbf{x}) - \langle \mathbf{y}, \mathbf{x}^\dagger - \mathbf{x}^\ddagger \rangle.$$

Combining the above two equalities, it holds that

$$
\begin{aligned}
&\mathrm{KL}(\mathbf{x}^*, \mathbf{x}^\ddagger) - \mathrm{KL}(\mathbf{x}^*, \mathbf{x}^\dagger) \\
&= -\mathrm{KL}(\mathbf{x}^\ddagger, \mathbf{x}^\dagger) - \mathrm{KL}(\mathbf{x}^\dagger, \mathbf{x}) + \langle \mathbf{y}^\dagger, \mathbf{x}^\ddagger - \mathbf{x}^* \rangle + \langle \mathbf{y}, \mathbf{x}^\dagger - \mathbf{x}^\ddagger \rangle \\
&= -\mathrm{KL}(\mathbf{x}^\ddagger, \mathbf{x}^\dagger) - \mathrm{KL}(\mathbf{x}^\dagger, \mathbf{x}) + \langle \mathbf{y}^\dagger, \mathbf{x}^\dagger - \mathbf{x}^* \rangle + \langle \mathbf{y}^\dagger - \mathbf{y}, \mathbf{x}^\ddagger - \mathbf{x}^\dagger \rangle \\
&\leq -\frac{1}{2} \|\mathbf{x}^\ddagger - \mathbf{x}^\dagger\|^2 - \frac{1}{2} \|\mathbf{x}^\dagger - \mathbf{x}\|^2 + \langle \mathbf{y}^\dagger, \mathbf{x}^\dagger - \mathbf{x}^* \rangle + \frac{1}{2} \|\mathbf{y}^\dagger - \mathbf{y}\|^2 + \frac{1}{2} \|\mathbf{x}^\ddagger - \mathbf{x}^\dagger\|^2 \\
&= \frac{1}{2} \|\mathbf{y}^\dagger - \mathbf{y}\|^2 - \frac{1}{2} \|\mathbf{x}^\dagger - \mathbf{x}\|^2 + \langle \mathbf{y}^\dagger, \mathbf{x}^\dagger - \mathbf{x}^* \rangle.
\end{aligned}
$$

Next, we estimate $\|\mathbf{y}^\dagger - \mathbf{y}\|^2$ and $\langle \mathbf{y}^\dagger, \mathbf{x}^\dagger - \mathbf{x}^* \rangle$. Recall the definition of $\mathbf{y}$ and $\mathbf{y}^\dagger$:

$$
\begin{aligned}
\mathbf{y} &= (\mathbf{y}_1^{n\mathcal{T}+i+j}, \mathbf{y}_2^{n\mathcal{T}+i+j}) \\
&= (\eta A_{n\mathcal{T}+i+j} \mathbf{x}_2^{n\mathcal{T}+i+j}, -\eta A_{n\mathcal{T}+i+j}^\top \mathbf{x}_1^{n\mathcal{T}+i+j}) \\
&= (\eta A_{i+j} \mathbf{x}_2^{i+j}, -\eta A_{i+j}^\top \mathbf{x}_1^{n\mathcal{T}+i+j}) \\
&= \eta \cdot \begin{bmatrix} & A_{i+j} \\ -A_{i+j}^\top & \end{bmatrix} \mathbf{x},
\end{aligned}
$$

and

$$
\begin{aligned}
\mathbf{y}^\dagger &= (\mathbf{y}_1^{n\mathcal{T}+i+j+\frac{1}{2}}, \mathbf{y}_2^{n\mathcal{T}+i+j+\frac{1}{2}}) \\
&= (\eta A_{n\mathcal{T}+i+j} \mathbf{x}_2^{n\mathcal{T}+i+j+\frac{1}{2}}, -\eta A_{n\mathcal{T}+i+j}^\top \mathbf{x}_1^{n\mathcal{T}+i+j+\frac{1}{2}}) \\
&= (\eta A_{i+j} \mathbf{x}_2^{n\mathcal{T}+i+j+\frac{1}{2}}, -\eta A_{i+j}^\top \mathbf{x}_1^{n\mathcal{T}+i+j+\frac{1}{2}}) \\
&= \eta \cdot \begin{bmatrix} & A_{i+j} \\ -A_{i+j}^\top & \end{bmatrix} \mathbf{x}^\dagger.
\end{aligned}
$$

Then we have that

$$\|\mathbf{y}^\dagger - \mathbf{y}\|^2 \leq \eta^2 \|A_{i+j}\|^2 \cdot \|\mathbf{x}^\dagger - \mathbf{x}\|.$$

and

$$\langle \mathbf{y}^\dagger, \mathbf{x}^\dagger - \mathbf{x}^* \rangle$$

$$= - (\mathbf{x}_1^*)^\top A_{i+j} \mathbf{x}_2^{n\mathcal{T}+i+j+\frac{1}{2}} + (\mathbf{x}_1^{n\mathcal{T}+i+j+\frac{1}{2}})^\top A_{i+j} \mathbf{x}_2^*$$

$$= (\mathbf{x}_1^*)^\top A_{i+j} \mathbf{x}_2^* - (\mathbf{x}_1^*)^\top A_{i+j} \mathbf{x}_2^{n\mathcal{T}+i+j+\frac{1}{2}} + (\mathbf{x}_1^{n\mathcal{T}+i+j+\frac{1}{2}})^\top A_{i+j} \mathbf{x}_2^* - (\mathbf{x}_1^*)^\top A_{i+j} \mathbf{x}_2^*$$

$$\leq 0,$$

where the last inequality comes from $\mathbf{x}_1$ is the maxima player, and $\mathbf{x}_2$ is the minima player.

Let $q = \max_{t \in [\mathcal{T}]} \|A_t\|$, then we have

$$\mathrm{KL}(\mathbf{x}^*, \mathbf{x}^\ddagger) - \mathrm{KL}(\mathbf{x}^*, \mathbf{x}^\dagger)$$

$$\leq \frac{1}{2}(\eta^2 q^2 - 1) \left\|\mathbf{x}^\dagger - \mathbf{x}\right\|^2 + \langle \mathbf{y}^\dagger, \mathbf{x}^\dagger - \mathbf{x}^* \rangle$$

$$\leq \frac{1}{2}(\eta^2 q^2 - 1) \left\|\mathbf{x}^\dagger - \mathbf{x}\right\|^2 < 0.$$

Then it can be concluded that

$$\mathrm{KL}\left((\mathbf{x}_1^*, \mathbf{x}_2^*), (\mathbf{x}_1^{n\mathcal{T}+i+j+1}, \mathbf{x}_2^{n\mathcal{T}+i+j+1})\right) - \mathrm{KL}\left((\mathbf{x}_1^*, \mathbf{x}_2^*), (\mathbf{x}_1^{n\mathcal{T}+i+j}, \mathbf{x}_2^{n\mathcal{T}+i+j})\right) < 0,$$

which leads to the result. $\qquad \square$

**Proposition B.13** (Discrete-time LaSalle invariance principle , LaSalle [1976])**.** *Let $G$ be any set in $\mathbb{R}^m$. Consider a difference equations system defined by a map $T : G \to G$ that is well defined for any $x \in G$ and continuous at any $x \in G$. Suppose there exists a scalar map $V : \bar{G} \to \mathbb{R}$ satisfying*

- *$V(x)$ is continuous at any $x \in \bar{G}$,*
- *$V(T(x)) - V(x) \leq 0$ for any $x \in G$.*

*For any $x_0 \in G$, if the solution to the following initial-value problem*

$$x(n+1) = T(x(n)), x(0) = x_0,$$

*satisfying that $\{x(n)\}_{n=1}^\infty$ is bounded and $x(n) \in G$ for any $n \in \mathbb{N}$, then there exists some $c \in \mathbb{R}$ such that $x(n) \to M \cap V^{-1}(c)$ as $n \to \infty$, where*

$$V^{-1}(c) = \{x \in \mathbb{R}^m | V(x) = c\},$$

*and $M$ is the largest invariant set in $E = \{x \in G \mid V(T(x)) - V(x) = 0\}$.*

**Proposition B.14.** *For any $i \in [\mathcal{T}]$ and $(\boldsymbol{x}_1^0, \boldsymbol{x}_2^0) \in \Delta_m \times \Delta_n$, we have*

$$\lim_{n \to \infty} \tilde{\mathcal{F}}_i^n \left((\boldsymbol{x}_1^0, \boldsymbol{x}_2^0)\right) = (\boldsymbol{x}_1^*, \boldsymbol{x}_2^*).$$

*Proof.* In Proposition B.13, we replace the dynamical system $T$ by $\tilde{\mathcal{F}}_i$ and the scalar map $V$ by the KL-divergence $\mathrm{KL}\left((\mathbf{x}_1^*, \mathbf{x}_2^*), \cdot\right)$. Note that as KL-divergence is defined as $+\infty$ on the boundary of simplex, thus $\mathrm{KL}\left((\mathbf{x}_1^*, \mathbf{x}_2^*), \cdot\right)$ is continuous function on the simplex.

From Proposition B.12, the invariant set $M$ can only the the single point set $\{(\mathbf{x}_1^*, \mathbf{x}_2^*)\}$, thus we have

$$\lim_{n \to \infty} \tilde{\mathcal{F}}_i^n \left((\mathbf{x}_1^0, \mathbf{x}_2^0)\right) = (\mathbf{x}_1^*, \mathbf{x}_2^*).$$

$\qquad \square$

The following Proposition character the attractor of a periodic dynamical system.

**Proposition B.15** (Theorem 3 in Franke and Selgrade [2003])**.** *Let $\Omega$ be an attractor for the $\mathcal{T}$-periodic dynamical system $f_0, f_1, ..., f_{\mathcal{T}-1}$. Then $\Omega = \cup_{i=0}^{\mathcal{T}-1} \Omega_i$, where $\Omega_i$ is an attractor for the map $f_{i+\mathcal{T}-1} \circ ... \circ f_i$, for $i \in [\mathcal{T}]$.*

Now Theorem 3.2 directly follows from Proposition 2.3, as it has been shown in our case $\Omega_i = \{(\mathbf{x}_1^*, \mathbf{x}_2^*)\}$.