# OpenReview forum: "Last-iterate Convergence Separation between Extra-gradient and Optimism in Constrained Periodic Games"
_auai.org/UAI/2024/Conference — UAI 2024 poster_

### Official Review · Reviewer_TLPM · 2024-02-27

**Q2-1 Originality-Novelty:** 2
**Q2-2 Correctness-Technical Quality:** 3
**Q2-5 Clarity Of Writing:** 3

**Q1 Summary And Contributions:**

This paper considers the problem of the difference between two well-known algorithms, extra-gradient (EG) and optimistic online gradient descent-ascent (OGDA). Both algorithms use historical information to improve the performance and exhibit similar convergence results in time-invariant games. This paper considers the natural problem of whether the two algorithms still behave similarly in time-varying games, a more practical scenario in non-stationary environments. This paper shows that in a special case of time-varying periodic games, OGDA (OMWU in matrix games) fails to converge, while EG (EG-MWU in matrix games) can still converge with bounded domains (simplexes). Empirical studies also validate the effectiveness of the convergence results.

**Q2-3 Extent To Which Claims Are Supported By Evidence:**

3: Good: the main claims are supported by convincing evidence (in the form of adequate experimental evaluation, proofs, (pseudo-)code, references, assumptions).

**Q2-4 Reproducibility:**

3: Good: key resources (e.g. proofs, code, data) are available and key details (e.g. proofs, experimental setup) are sufficiently well-described for competent researchers to confidently reproduce the main results.

**Q3 Main Strengths:**

This paper considers the meaningful problem of the difference between OGDA and EG, two well-known algorithms in the game theory, and shows their difference in the time-varying case, which is valuable. The presentation is clear, intuitive, and thus easy to follow. The empirical studies are also sufficient as evidence for the theoretical guarantees.

**Q4 Main Weakness:**

I have read the paper of NeurIPS 2023 (On the last-iterate convergence in time-varying zero-sum games: Extra gradient succeeds where optimism fails), which is an excellent work in finding the difference between these two algorithms in **unbounded** time-varying periodic games. As a result, when I read the title of this work of the first sight, I am curious about what is the key difference or challenge when adapting the results of the NeurIPS 2023 paper to bounded domains. However, I did not find such statements in the main paper. And I have a brief check of Theorem 3.2 (the convergence of EG-MWU), and I did not find anything about the challenges and difficulties when doing this in bounded domains either. Maybe I have missed this point and I am happy that the authors could give me some further explanations. On the other hand, the current writing of this work can be still improved and the authors could revise the paper to give more explanations on the challenges and difficulties in bounded domains. As a result, I am inclined to reject this paper and another round of review after the revision would be better. The current version makes me feel that adapting to bounded domains is just a natural extension of the NeurIPS 2023 paper, which makes the overall contributions of this work insufficient for publication.

**Q5 Detailed Comments To The Authors:**

Questions:
1. The authors mentioned that handling time-varying periodic games is challenging since the system is non-autonomous and non-linear. In Section 2.3, the authors introduced two methods that can transform the system into an autonomous and local linear one. Are the authors trying to say that the time-varying periodic game is not that hard since there exist some reduction methods?
2. In the first paragraph of the introduction, the author said, "It is well-known that these learning algorithms have no-regret property, ...". However, I remember that the EG algorithm is not a regret-minimization method. I hope that the authors could have a double check on this point.

**Q9 Complying With Reviewing Instructions:**

Yes

---

> ### Author Rebuttal · Authors · 2024-04-05
>
> We thank the reviewer for the careful reading and constructive comments. We will address your questions and we will be always open for further discussion. Please see our itemized responses below:
>
> **1.Technical comparison between the current work and [Feng et al., 2023].**
>
> Thank you for pointing out this aspect. We will definitely include the following section in the revised version to provide a more detailed technical comparison between the current work and [Feng et al., 2023].
>
>
> > ***Technical Comparison.***  The MWU-based algorithms considered in this paper differ from the GDA-based algorithms considered in [Feng et al., 2023] in two fundamental ways. Firstly, variations of MWU algorithms are naturally defined on the simplex constraints, allowing our analysis to avoid the difficulty of projecting onto simplex. Secondly, the algorithms considered in [Feng et al., 2023] have linear structure, i.e., can be directly analyzed as linear systems, while the MWU-based algorithms have  **non-linear**, making the techniques of [Feng et al., 2023] ineffective in our scenario. At a high level, by considering variations of MWU algorithms, we transform the technical difficulties arising from constraints into difficulties related to analyzing non-linear dynamics of MWU-based algorithms. It is worth noting that a similar transformation can be observed in the line of research on establishing last-iterate convergence in static games: [1] first proved convergence of Optimistic GDA without constraints and then [2] extended their results to constrained settings for Optimistic MWU.
> >
> >\
> > To be more specific, allow us to compare the proof of Theorem 3.2 in our paper with Theorem 3.1 in [Feng et al., 2023]. Note that both theorems demonstrate that Extra-gradient methods converge in periodic games.
>  > In [Feng et al., 2023], the authors rely on a key fact that characterizes the iterative matrices of Extra-GDA as normal matrices with eigenvalues not exceeding 1 (Lemma A.2 and B.1 in their paper). This simplifies the problem by leveraging existing tools in linear dynamical systems. A natural extension of their approach to Extra-MWU algorithms would be to prove similar results for the *Jacobian* matrix of Extra-MWU near the equilibrium and as a result provide guarantees in a small neighborhood around the equilibrium. Nevertheless, this does not imply the **global** convergence property of Extra-MWU in periodic games, due to its non-linearity; the *Jacobian* matrix can only describe behaviors of points close to equilibrium, resulting in a **local** convergence result for Extra-MWU (local guarantees). To obtain a global convergence result, we employ a Lyapunov-type argument specifically tailored for Extra-MWU (Section 4.2 in our paper), and this differs significantly from [Feng et al., 2023]. A similar difficulty also appears in establishing the divergence result of Optimistic MWU.
>
> ---
> **2.Are the authors trying to say that the time-varying periodic game is not that hard since there exist some reduction methods?**
>
> Yes, we agree that periodic games are a suitable first step beyond static games, as they provide sufficient technical challenges (please refer to the technical comparison paragraph) and nontrivial phenomena, while still being tractable, towards understanding the learning dynamics in more general time-varying games, as demonstrated by several previous works such as [3] and [4].
>
> ---
> **3.Regret property of EG methods.**
>
> Thank you for pointing this out. EG is not a no-regret learning algorithm. We will fix this in the revised version.
>
> References:
>
> [1] Daskalakis et al., Training GANs with Optimism, ICLR 2018
>
> [2] Daskalakis and Panageas, Last-iterate convergence: Zero-sum games and constrained min-max optimization, ITCS 2019
>
> [3] Fiez et al., Online Learning in Periodic Zero-Sum Games, NeurIPS 2021
>
> [4] Feng et al., On the Last-iterate Convergence in Time-varying Zero-sum Games: Extra Gradient Succeeds where Optimism Fails, NeurIPS 2023

---

### Official Review · Reviewer_KA9W · 2024-03-18

**Q2-1 Originality-Novelty:** 2
**Q2-2 Correctness-Technical Quality:** 3
**Q2-5 Clarity Of Writing:** 3

**Q1 Summary And Contributions:**

This paper investigates the last-iterate convergence behavior of two families of algorithms: extra-gradient methods and optimistic methods, with a particular focus on periodic games. The authors demonstrate that, within a constrained setting, there are specific challenging instances where extra-gradient methods can achieve convergence, whereas optimistic methods fail. This outcome is established by modeling the learning process through dynamical systems. Additionally, the paper outlines conditions under which optimistic methods can achieve convergence within the given setting. The findings are validated by a series of numerical experiments.

**Q2-3 Extent To Which Claims Are Supported By Evidence:**

3: Good: the main claims are supported by convincing evidence (in the form of adequate experimental evaluation, proofs, (pseudo-)code, references, assumptions).

**Q2-4 Reproducibility:**

2: Fair: key resources (e.g. proofs, code, data) are unavailable but key details (e.g. proof sketches, experimental setup) are sufficiently well-described for an expert to confidently reproduce the main results.

**Q3 Main Strengths:**

This paper aims to provide a more detailed theoretical description for the seperation for two classic methods: the extra-gradient methods and the optimistic methods. The authors further identify sufficient conditions that enable convergence of optimistic methods, thereby enriching the theoretical landscape for constrained periodic games. The numetrical experiments supports their findings.

**Q4 Main Weakness:**

The paper lacks a discussion on the challenges encountered while proving the results within the constrained domain. It would be beneficial to offer more intuitive explanations for achieving Propositions 4.1 to 4.2, and highlight the improvements/challenges compared to the previous works.

**Q5 Detailed Comments To The Authors:**

- There is inconsistency in the use of punctuation within the proof sections, with some equations lacking the requisite commas or periods.
- There is a typo for the equation at the bottom of page 20.
- Could the authors provide more detailed discussions about the technical challenges when proving the results within the constrained domain?

**Q9 Complying With Reviewing Instructions:**

Yes

---

> ### Author Rebuttal · Authors · 2024-04-05
>
> We thank the reviewer for the careful reading and constructive comments. We will address your questions and we will be always open for further discussion. Please see our itemized responses below:
>
> **1.Typos and inconsistency in the use of punctuation.**
>
> Thank you for pointing this out! We will definitely correct any typos and punctuation errors in the revised version.
>
> ---
>
> **2.More detailed discussions about the technical challenges when proving the results within the constrained domain.**
>
>  We provide the same answer as to reviewer TLPM. We will add the following paragraph as a section on technical challenges and comparison in the revised version.
>
> > ***Technical Comparison.***  The MWU-based algorithms considered in this paper differ from the GDA-based algorithms considered in [Feng et al., 2023] in two fundamental ways. Firstly, variations of MWU algorithms are naturally defined on the simplex constraints, allowing our analysis to avoid the difficulty of projecting onto simplex. Secondly, the algorithms considered in [Feng et al., 2023] have linear structure, i.e., can be directly analyzed as linear systems, while the MWU-based algorithms have  **non-linear**, making the techniques of [Feng et al., 2023] ineffective in our scenario. At a high level, by considering variations of MWU algorithms, we transform the technical difficulties arising from constraints into difficulties related to analyzing non-linear dynamics of MWU-based algorithms. It is worth noting that a similar transformation can be observed in the line of research on establishing last-iterate convergence in static games: [1] first proved convergence of Optimistic GDA without constraints and then [2] extended their results to constrained settings for Optimistic MWU.
> >
> >\
> > To be more specific, allow us to compare the proof of Theorem 3.2 in our paper with Theorem 3.1 in [Feng et al., 2023]. Note that both theorems demonstrate that Extra-gradient methods converge in periodic games.
>  > In [Feng et al., 2023], the authors rely on a key fact that characterizes the iterative matrices of Extra-GDA as normal matrices with eigenvalues not exceeding 1 (Lemma A.2 and B.1 in their paper). This simplifies the problem by leveraging existing tools in linear dynamical systems. A natural extension of their approach to Extra-MWU algorithms would be to prove similar results for the *Jacobian* matrix of Extra-MWU near the equilibrium and as a result provide guarantees in a small neighborhood around the equilibrium. Nevertheless, this does not imply the **global** convergence property of Extra-MWU in periodic games, due to its non-linearity; the Jacobian matrix can only describe behaviors of points close to equilibrium, resulting in a **local** convergence result for Extra-MWU (local guarantees). To obtain a global convergence result, we employ a Lyapunov-type argument specifically tailored for Extra-MWU (Section 4.2 in our paper), and this differs significantly from [Feng et al., 2023]. A similar difficulty also appears in establishing the divergence result of Optimistic MWU.
>
> References:
>
> [1] Daskalakis et al., Training GANs with Optimism, ICLR 2018
>
> [2] Daskalakis and Panageas, Last-iterate convergence: Zero-sum games and constrained min-max optimization, ITCS 2019
>
> [3] Feng et al., On the Last-iterate Convergence in Time-varying Zero-sum Games: Extra Gradient Succeeds where Optimism Fails, NeurIPS 2023

---

### Official Review · Reviewer_vTvY · 2024-03-23

**Q2-1 Originality-Novelty:** 2
**Q2-2 Correctness-Technical Quality:** 3
**Q2-5 Clarity Of Writing:** 3

**Q1 Summary And Contributions:**

The paper focuses on analyzing the last-iterate behaviors of learning algorithms in repeated two-player zero-sum games. The motivation behind this study stems from the observation that most existing results on the last-iterate convergence of optimistic and extra-gradient methods have been established under the assumption that the game is time-independent. This paper seeks to extend the understanding of these behaviors to time-varying (specifically, periodic) games, which are more common in both practical and theoretical studies.

Key contributions of this paper include:

The demonstration of a clear last-iterate convergence separation between optimistic multiplicative weights update (OMWU) and extra-gradient multiplicative weights update (Extra-MWU) methods in constrained periodic games. This separation challenges the conventional wisdom that these two methods would behave similarly, as they do in time-independent games.

The construction of a constrained periodic game with a common equilibrium, where it's proven that OMWU does not converge to the equilibrium, highlighting a novel finding that contrasts with previous understandings of learning algorithms' behavior in static games.

The proof that Extra-MWU will converge to the common equilibrium in periodic games with simplex constraints, thereby providing a theoretical underpinning for the application of Extra-MWU in more complex, time-varying game scenarios.

Extensive numerical experiments to support the theoretical findings, offering insights into the practical implications and potential applications of these convergence behaviors in various machine learning contexts.

**Q2-3 Extent To Which Claims Are Supported By Evidence:**

3: Good: the main claims are supported by convincing evidence (in the form of adequate experimental evaluation, proofs, (pseudo-)code, references, assumptions).

**Q2-4 Reproducibility:**

3: Good: key resources (e.g. proofs, code, data) are available and key details (e.g. proofs, experimental setup) are sufficiently well-described for competent researchers to confidently reproduce the main results.

**Q3 Main Strengths:**

1. It confirms that in Constrained Periodic Games, OMWU and Extra-MWU exhibit different behaviors both theoretically and experimentally. Specifically, OMWU diverges without reaching equilibrium, while Extra-MWU achieves last-iterate convergence.

2. This paper tackles challenges previously listed as future work by [Feng+ 2023], providing solutions and advancing the field.

**Q4 Main Weakness:**

The weakness of this study is that it only presents asymptotic results, without deriving the convergence rate.

**Q5 Detailed Comments To The Authors:**

1. In exploring the dynamics of constrained periodic games, have you considered the potential results of applying Optimistic Gradient Descent-Ascent (OGDA) and Extra-gradient Gradient Descent-Ascent (extra-GDA) learning algorithms to these games? I understand that projecting strategies onto the simplex is a necessity in this context. It would be interesting to hear your predictions or any preliminary insights you might have on this matter.

2. Given the emphasis on specific learning dynamics in your study, do you envisage the possibility of extending your findings to more general learning algorithms, like Online Mirror Descent (OMD), which include both Gradient Descent-Ascent (GDA) and Multiplicative Weights Update (MWU)?

(Optional)
3. On a related note—driven by curiosity rather than a suggestion for manuscript improvement—I'm interested in your intuitive explanation for why OGDA might struggle in learning in these settings, whereas extra-GDA could achieve LIC. Is there a straightforward rationale behind the differential performance of these algorithms in constrained periodic game scenarios?

**Q9 Complying With Reviewing Instructions:**

Yes

---

> ### Author Rebuttal · Authors · 2024-04-05
>
> We thank the reviewer for the careful reading and constructive comments. We will address your questions and we will be always open for further discussion. Please see our itemized responses below:
>
> **1.The potential results of applying Optimistic Gradient Descent-Ascent (OGDA) and Extra-gradient Gradient Descent-Ascent (extra-GDA) learning algorithms to these games.**
>
> We believe the same last-iterate separation behaviors between the projected OGDA and Extra-GDA  algorithms should also exist. We verify this through some numerical experiments, and we present the numerical results and possible explanations about this point in the following anonymous link :
>
> `https://www.dropbox.com/scl/fi/aqrf1t3fuqdu8j409lq28/rebuttle_periodic_OMWU_and_ExtraMWU.pdf?rlkey=rjdb20jws5gqe6fyv0dqaya4q&dl=0`
>
> We agree that provide a rigorous proof of these observations is an interesting future work.
>
> ---
> **2.Possibility of extending the findings to more general learning algorithms, like Online Mirror Descent (OMD).**
>
> Thank you for your questions regarding the extension of the results in the current paper to other learning algorithms. We believe that the results here can also hold for general Optimistic variants and Extra-gradient variants of Online Mirror Descent algorithms. For Optimistic methods, based on the result in the current paper and [1], it seems that their divergence to the boundary behaviors (since [1] considers an unbounded setting, thus their algorithm's dynamics indeed diverge to infinity) are robust under changes in constraints. As for Extra-gradient methods, the key point is to show that the behaviors of Bregman divergence are similar to those of KL-divergence in Extra-gradient MWU. We believe this should also hold since several important tools such as three-points identity and convex dual properties for Extra-gradient MWU also apply to general Extra-gradient Mirror Descent. We believe extending the findings to more general learning algorithms is an interesting and promising future direction.
>
> ---
> **3.Is there a straightforward rationale behind the differential performance of these algorithms in constrained periodic game scenarios?**
>
> Our answer is similar to that of reviewer 1oNs. We provide the following explanations as a high-level idea about the separation on the dynamical behaviors between Optimistic and Extra-gradient methods in periodic games : In general, Optimistic methods use the idea of "predication" to achieve regret minimization and convergence (for example, see [2] and [3]). In the example as we created in Theorem 3.1, we have $A_t = -A_{t+1}$ for the payoff matrix in two consecutive rounds in the repeated game. This instability causes the prediction failure and leads to the divergence of Optimistic MWU. However, Extra-gradient methods follow a different idea of "extrapolation" (See [4]) to achieve convergence. It seems that the idea of extrapolation makes the algorithm more robust under changes in the underlying payoff matrices. A formal proof of this point is provided in the proof of our Theorem 3.2, where we show that even when the underlying payoff matrices can change, the KL-divergence between the current strategies and the equilibrium steadily declines, leading to the convergence of Extra-gradient MWU.
>
> ---
> **4.This study is that it only presents asymptotic results, without deriving the convergence rate.**
>
> Thanks for your comments. We agree that provide concrete rate of these online learning algorithms in time-varying games is an interesting future direction. One idea would be to provide refined analysis of our Lyapunov arguments so that rates can be derived.
>
> References:
>
> [1] Feng et al., On the Last-iterate Convergence in Time-varying Zero-sum Games: Extra Gradient Succeeds where Optimism Fails, NeurIPS 2023
>
> [2] Alexander Rakhlin and Karthik Sridharan, Optimization, Learning, and Games with Predictable Sequences, NeurIPS 2013
>
> [3] Daskalakis et al., Training GANs with Optimism, ICLR 2018
>
> [4] Korpelevich, The Extragradient Method for Finding Saddle Points and Other Problems, Matecon, 12:747–756, 1976.

---

### Official Review · Reviewer_1oNs · 2024-03-23

**Q2-1 Originality-Novelty:** 3
**Q2-2 Correctness-Technical Quality:** 3
**Q2-5 Clarity Of Writing:** 3

**Q1 Summary And Contributions:**

The paper studies a constrained time-varying zero-sum game and shows that while extra OMWU may converge in last iterate, vanilla MWU does not.

**Q2-3 Extent To Which Claims Are Supported By Evidence:**

3: Good: the main claims are supported by convincing evidence (in the form of adequate experimental evaluation, proofs, (pseudo-)code, references, assumptions).

**Q2-4 Reproducibility:**

3: Good: key resources (e.g. proofs, code, data) are available and key details (e.g. proofs, experimental setup) are sufficiently well-described for competent researchers to confidently reproduce the main results.

**Q3 Main Strengths:**

1. The last-iterate property in a constrained zero-sum game is very interesting.
2. The result on extra MWU and OMWU is surprising.
3. The theoretical result is complemented by detailed empirical result.

**Q4 Main Weakness:**

I would suggest more explanation on the high-level idea of why would OMWU fail while extra MWU succeeded. To my understanding, both algorithms can be seen as approximations of the proximal gradient algorithm. Also, the constraint on the action set is restricted to simplex, I am not sure if the result can be applied to other constrained action sets.

**Q5 Detailed Comments To The Authors:**

1. Could you elaborate on the high-level idea of why would OMWU fail while extra MWU succeeded?
2. Can the result be extended to other constrained action set?

**Q9 Complying With Reviewing Instructions:**

Yes

---

> ### Author Rebuttal · Authors · 2024-04-05
>
> We thank the reviewer for the careful reading and constructive comments. We will address your questions and we will be always open for further discussion. Please see our itemized responses below:
>
> **1.High-level idea of why would OMWU fail while extra MWU succeeded.**
>
> There are intuitive explanations for this phenomenon. Firstly, Optimistic methods use the idea of "predication" to achieve regret minimization and convergence (for example, see [1] and [2]). However, in the example as we created in Theorem 3.1, we have $A_t = -A_{t+1}$ for the payoff matrix in two consecutive rounds in the repeated game. This instability causes the prediction failure and leads to the divergence of Optimistic MWU. Secondly, Extra-gradient methods follow a different idea of "extrapolation" (See [3]) to achieve convergence. It seems that the idea of extrapolation makes the algorithm more robust under changes in the underlying payoff matrices. A formal proof of this point is provided in the proof of our Theorem 3.2, where we show that even when the underlying payoff matrices can change, the KL-divergence between the current strategies and the equilibrium steadily declines, leading to the convergence of Extra-gradient MWU.
>
> ---
> **2. Can the results be extended to other constrained action sets?**
>
> Thank you for your questions regarding the extension of the results in the current paper to other constrained settings. We believe that our findings can also be applied to other action sets with constraints. Based on our results and those in [4], it appears that the divergence behaviors of Optimistic methods remain robust under different constraints. As for Extra-gradient methods, we expect that general variants of Follow-the-Regularized-Leader algorithms can achieve convergence to the last iterate in periodic games, as some structures such as Jacobian matrices of these algorithms are similar to Extra-gradient MWU (for example, see [5]). We consider extending these current results to other settings as an interesting and promising future direction.
>
> References:
>
> [1] Alexander Rakhlin and Karthik Sridharan, Optimization, Learning, and Games with Predictable Sequences, NeurIPS 2013
>
> [2] Daskalakis et al., Training GANs with Optimism, ICLR 2018
>
> [3] Korpelevich, The Extragradient Method for Finding
> Saddle Points and Other Problems, Matecon, 12:747–756, 1976.
>
> [4] Feng et al., On the Last-iterate Convergence in Time-varying Zero-sum Games: Extra Gradient Succeeds where Optimism Fails, NeurIPS 2023
>
> [5] Yun Kuen Cheung and Georgios Piliouras, Vortices Instead of Equilibria in MinMax Optimization: Chaos and Butterfly Effects of Online Learning in Zero-Sum Games, COLT 2019

---

### Meta-Review · Area_Chair_3HfZ · 2024-04-17

The authors show that the last-iterate behavior of learning algorithms in repeated *periodic* two-player zero-sum games can be significantly different between the optimistic and extragradient algorithms in constrained settings on the simplex. Most reviewers are quite positive about the paper, and the authors did a good job explaining non-triviality of their proof techniques over the paper of (Feng et al, 2023) which studied the unconstrained case.